# Regulatory networks of KRAB zinc finger genes and transposable elements changed during human brain evolution and disease

Yao-Chung Chen[1,2]*, Arnaud Maupas[3], Katja Nowick[1,2]*

[1]Human Biology and Primate Evolution, Institute of Biology, Freie Universität Berlin, Berlin, Germany; [2]Institute of Bioinformatics, Freie Universität Berlin, Berlin, Germany; [3]Institut de Biologie de l'École Normale Supérieure, Ecole Normale supérieure de Paris, Paris, France

**\*For correspondence:**
yao-chung.chen@fu-berlin.de (Y-CC);
katja.nowick@fu-berlin.de (KN)

**Competing interest:** The authors declare that no competing interests exist.

## eLife Assessment

The authors present a software (TEKRABber) to analyze how expression of transposable elements (TEs) and TE silencing factors KRAB zinc finger (KRAB-ZNF) genes are correlated in experimentally validated datasets. TEKRABber is used to reconstruct regulatory networks of KRAB-ZNFs and TEs during human brain evolution and in Alzheimer's disease. The direction of the work is **important**, with potentially significant interest from others looking for a tool for correlative gene expression analysis across individual genomes and species. However, the reviews identified biases and shortcomings in the pipeline that could lead to an unacceptable number of false positive and negative signals and thus impact the conclusions, leaving the work in its current form **incomplete**.

**Abstract** Evidence indicates that transposable elements (TEs) can contribute to the evolution of new traits, with some TEs acting as deleterious elements while others are repurposed for beneficial roles in evolution. In mammals, some KRAB-ZNF proteins can serve as a key defense mechanism to repress TEs, offering genomic protection. Notably, the family of KRAB-ZNF genes evolves rapidly and exhibits diverse expression patterns in primate brains, where some TEs, including autonomous LINE-1 and non-autonomous Alu and SVA elements, remain mobile. This prompts questions about their interactions in primate brains and potential roles in human brain evolution and disease. For a systematic comparative analysis of TE interactions with other genes, we developed the tool TEKRABber and focused on strong and experimentally validated cases. Our bipartite network analysis revealed significantly more interactions between KRAB-ZNF genes and TEs in humans than in other primates, especially with recently evolved, i.e., Simiiformes-specific, TEs. Notably, ZNF528, under positive selection in humans, shows numerous human-specific TE interactions. Most negative interactions in our network, indicative of repression by KRAB-ZNF proteins, entail Alu TEs, while links to other TEs are generally positive. In Alzheimer's patients, a subnetwork involving 21 interactions with an Alu module appears diminished or lost. Our findings suggest that KRAB-ZNF and TE interactions vary across TE families, have increased throughout human evolution, and may influence susceptibility to Alzheimer's disease.

## Introduction

Transposable elements (TEs) are repetitive DNA sequences capable of migrating and inserting into new locations within the host genome. When transposed, TEs may induce phenotypic changes in the organism (*Bourque et al., 2018*; *Schrader and Schmitz, 2019*). One well-known textbook example is the development of an industrial melanization mutant phenotype in the British peppered moth, *Biston betularia*, which has been linked to increased expression of the *cort* transcript due to a TE insertion in its first intron (*van Hof et al., 2016*). Another classic example for primate evolution is the retrovirus insertion in the salivary amylase gene, *AMY1C*, with this sequence being essential for primates to evolve as one of the few mammals capable of expressing amylase in saliva (*Ting et al., 1992*). A further example is the evolutionary loss of tails in humans and apes, which is associated with the insertion of an Alu retrotransposon into the intronic region of the *TBXT* gene. This insertion creates an alternative splice site with an ancestral Alu retrotransposon in the sequence, leading to the excision of the middle exon and the formation of an isoform (*Xia et al., 2024*).

TEs comprise about half or even more of the mammalian genome (*Platt et al., 2018*; *Qu et al., 2023*). Their positions in the genome can be annotated with tools like RepeatMasker (*Lawson et al., 2023*). TE insertions can be deleterious, such as the insertion of an L1 element initiating colorectal cancer in humans (*Scott et al., 2016*). Therefore, a variety of host factors are employed to regulate TE expression, such as small RNAs, chromatin and DNA modification pathways, and KRAB zinc finger (KRAB-ZNF) proteins (*Colonna Romano and Fanti, 2022*). KRAB-ZNF proteins are the largest family of transcription factors in higher vertebrates (*Huntley et al., 2006*; *Ecco et al., 2017*), characterized by fast evolution and contributing to gene expression differences between primates (*Nowick et al., 2009*; *Nowick et al., 2013*). KRAB-ZNF proteins bind to the interspersed DNA sequences of TEs and repress their expression upon recruiting the cofactor KAP1 (*Groner et al., 2010*). Fast evolution was also reported for TEs, which led to hypothesizing an evolutionary arms-race model, in which mutated TEs have the chance to escape the repression from KRAB-ZNF proteins until the KRAB-ZNF proteins evolve again with a suitable recognition ability (*Jacobs et al., 2014*; *Imbeault et al., 2017*). For instance, in primates, coevolution between a family of TEs, retroelements in endogenous retroviruses (ERVs), and the tandem repeats in KRAB-ZNF genes has been observed (*Thomas and Schneider, 2011*).

Studying the expression of KRAB-ZNF genes and TEs in the evolution of the human brain has crucial relevance, since some TEs are actively transcribed in the developing human brain (*Bodea et al., 2018*), including L1 subfamilies, HERV-K subfamilies, and primate-specific Alu subfamilies (*Hancks and Kazazian, 2010*; *Larsen et al., 2018*; *Dembny et al., 2020*). Furthermore, TE-derived promoters play a role in gene transcription within the human brain, which in turn suggests the significance of TEs in gene regulation during neurodevelopment (*Playfoot et al., 2021*). Interestingly, a substantial number of differentially expressed primate-specific KRAB-ZNF genes were detected in the adult prefrontal cortex, comparing humans to chimpanzees (*Nowick et al., 2009*), and newly evolved KRAB-ZNF genes were predominantly detected in the developing human brain (*Zhang et al., 2011*). Therefore, analyzing the interplay between TEs and KRAB-ZNF genes in the context of primate brain evolution could provide valuable insights into the complex mechanisms shaping human brain development and function.

The dysregulation of TE expression has also been linked to neurodegenerative diseases, including Alzheimer's disease (AD), the primary cause of dementia (*Ravel-Godreuil et al., 2021*). One of the biomarkers of AD, the Tau protein, appears to induce the expression of at least some TEs (*Guo et al., 2018*), and the presence of TE products in the cytosol and endosomes induces neuroinflammation in AD patients (*Evering et al., 2023*). Previous studies have suggested that cognitive skills are related to zinc finger genes and proteins. For instance, patients carrying the schizophrenia-risk allele of *ZNF804A* showed differences in their reading and spelling performance (*Becker et al., 2012*). Additionally, the single nucleotide polymorphism of a KRAB-ZNF gene, *ZNF224*, is associated with AD neuropathology and cognitive functions (*Shulman et al., 2010*).

Taken together, the study of the regulatory networks involving KRAB-ZNF genes and TEs is expected to provide insights into the evolution of the human brain and the development of neurodegenerative diseases. Here, we present results from two independent RNA-seq datasets: one comparing different brain regions between humans and multiple nonhuman primates (NHPs) (Primate Brain Data) and the other comparing human control samples with AD samples (Mayo Data). These datasets collectively

encompass a total of 514 samples from different species, brain regions, and disease states. Since it is very challenging to quantify and normalize expression of TEs for across-species comparison, we developed the TEKRABber R package for the systematic and comparative analysis of TE subfamilies (abbreviated as TEs in the following content). TEKRABber can further be used to explore expression correlations between TEs and any genes in any species of interest. Here, we used it to explore correlations with KRAB-ZNF genes in primates. Our work reveals an intricate network of TEs and KRAB-ZNF genes in the brain that changed during evolution and is modified in AD brains.

## Results

### TEKRABber: a software for cross-species comparative analysis of orthologs, TEs, and their co-expression

While computational tools for the analysis of TE expression in samples of a given species have already been developed (*Table 1*), to our knowledge, no tool exists yet that can compare TE expression across species, hampering the investigation of the impact of TEs on species evolution. Whereas the expression of orthologous genes can be compared between closely related species relatively easily, this task is more challenging for TEs, because the same TEs can be located in different regions on chromosomes and have different sequence lengths. Additionally, differences in the copy number of TEs between species can further complicate these comparisons. To gain functional evolutionary insights, it is further desired to estimate pairwise correlations between TEs and genes, which can be used to derive and compare regulatory networks involving TEs across species. We are aware of one method, TEffectR (*Karakülah et al., 2019*), that by providing mapped BAM files and specific locus regions, can be used for calculating TE and gene expression. Using a linear regression model with TE expression values, it subsequently predicts the impact of the TE on proximal gene expression in one species. However, it is not designed for contrasting correlations between pairwise orthologous genes and TEs across species directly from RNA-seq expression data. To better enable evolutionary studies on TEs, we developed an R Bioconductor software called TEKRABber (DOI: 10.18129/B9.bioc.TEKRABber). As a first use case for this new software, we investigated the interplay between TEs and KRAB-ZNF proteins, which can repress TE expression; hence the name TEKRABber. In a broader scope, TEKRABber addresses two primary challenges: comparing TE expression across species and efficiently calculating pairwise correlations between selected orthologous genes and TEs. With these features, it provides functionality not yet implemented in other tools (*Table 1*). It can also be used for exploring correlations of TEs with any other genes in any other species with genomes with TE annotations.

TEKRABber is designed to handle various types of transcriptomic read counts and offers two distinct modes of analysis (*Figure 1A*). In the first mode, tailored for interspecies comparison, we

**Table 1.** Comparison of transposable element (TE) expression analysis software.

| Software name | Description | Comparison feature | References |
|---|---|---|---|
| RepEnrich | Combines different mapping strategies for differentially expressed TE analysis using RNA-seq and ChIP-seq data | Different conditions (same species) | *Criscione et al., 2014* |
| TETools | Compares TE expression from RNA-seq data | Different conditions (same species) | *Lerat et al., 2016* |
| Telescope | Estimates TEs in specific genomic locations using RNA-seq data | One condition in one species | *Bendall et al., 2019* |
| TE Density | Provides a metric showing the presence of TEs relative to genes within flexible genomic distance | One condition in one species | *Teresi et al., 2022* |
| PlanTEenrichment | Calculates TE enrichment upon inputting a differentially expressed gene list and selection of a specific plant species | Different conditions (same species) | *Eskier et al., 2023* |
| GeneTEFlow | A nextflow pipeline for analyzing differential expression of genes and TEs | Different conditions (same species) | *Liu et al., 2020* |
| TEffectR | Estimates the proximal TE effects on gene expression using a linear regression model | Different conditions (same species) | *Karakülah et al., 2019* |
| TEKRABber | Computes differentially expressed genes/TEs and one-to-one correlations using RNA-seq data | Different conditions (same species) Across species comparison (different species) | Method presented here |

utilized the **Primate Brain Data** as a demonstration. Initially, it retrieved annotations from Ensembl (*Harrison et al., 2024*) for orthologs and from RepeatMasker (*Smit et al., 2013*) for TEs to estimate normalizing factors, ensuring comparable expression levels between species. This approach minimizes the likelihood that differences in fold change for differential expression (DE) are caused by TE or gene length variations. It also guarantees that only orthologs and TEs with high orthology confidence are included in the comparison, avoiding bias toward any particular species (*Figure 1—figure supplement 1*). Subsequently, users can employ the output data object to conduct DE analysis and identify one-to-one correlations based on selected parameters. We demonstrate that the impact of scaling is most pronounced for comparisons between the most distantly related species in our study, with about 30% of TEs being detected as DE or not between humans and rhesus macaques, depending on whether the expression data were scaled or not for the comparison (*Figure 1—figure supplement 1*). The second mode is designed for comparing different conditions, such as control and disease states within the same species. In this scenario, we used the **Mayo Data** as an example. Users can bypass the interspecies normalization steps and directly generate data objects for DE and correlation analyses. Notably, TEKRABber (from version 1.8, Bioc3.19) includes a parallel computing option, significantly enhancing computational efficiency based on the number of cores a device can provide. Furthermore, TEKRABber offers an interactive user interface, providing users with an initial overview of their results before delving into the details (*Figure 1B*).

In our study, we used TEKRABber to explore the putative functional connections between KRAB-ZNFs and TEs in the context of human brain evolution and AD. We conducted an analysis of KRAB-ZNF genes and TEs expression patterns and networks using two independent RNA-seq datasets: **Primate Brain Data** and **Mayo Data** (*Figure 1A*, *Supplementary file 1, tables S1–S3*). The Primate Brain

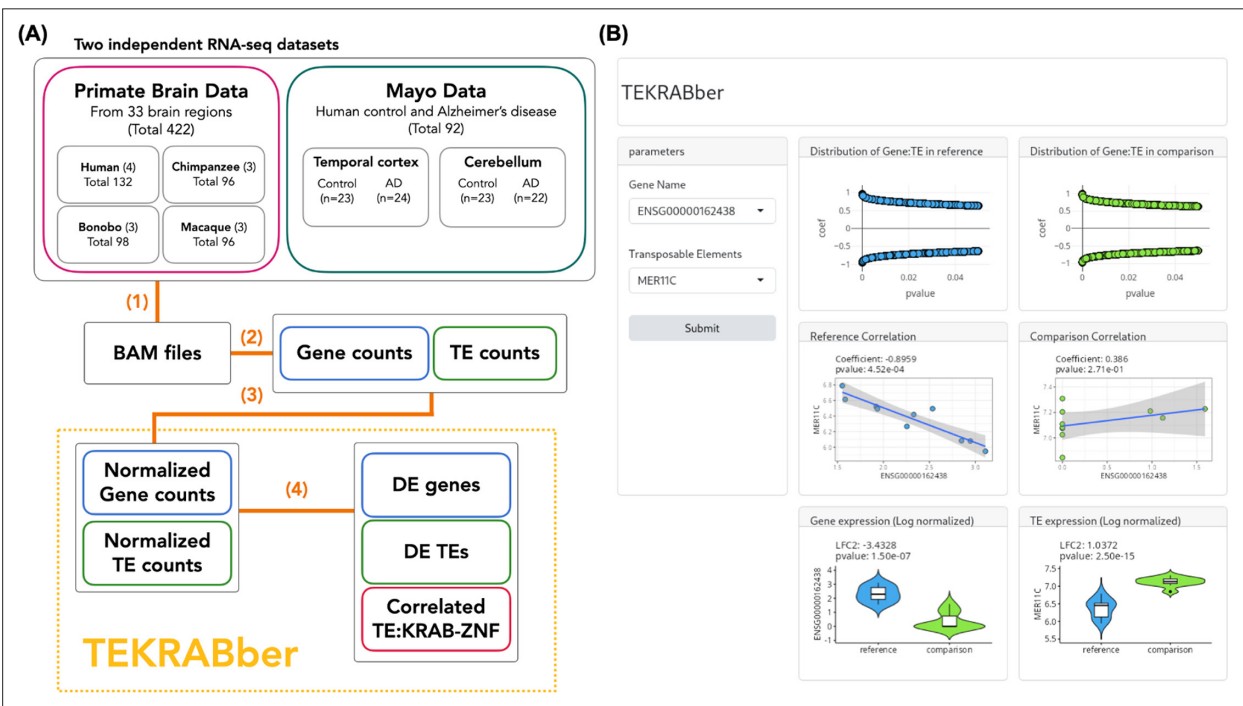

**Figure 1.** TEKRABber and the overview of the analysis workflow. (**A**) Two independent RNA-seq datasets, Primate Brain Data and Mayo Data, were analyzed in this study. (1) Transcriptomic data were first preprocessed by removing adapters and low-quality reads and then mapped to their reference genome using STAR to generate BAM files. (2) TEtranscript was used to quantify the expression of genes and transposable elements (TEs). (3) Expression profiles were normalized across different species. (4) Differential expression (DE) analysis and pairwise correlations were calculated. Steps (3) and (4) were developed together in an R Bioconductor package, TEKRABber. (**B**) The user interface of TEKRABber features a dashboard layout that allows users to explore one-to-one gene-TE interactions, including correlation and differential expression results (more details in Materials and methods section).

The online version of this article includes the following figure supplement(s) for figure 1:

**Figure supplement 1.** Comparison of differentially expressed (DE) transposable elements (TEs) with and without scaling.

**Figure supplement 2.** Graphical abstract of the analysis.

Data contains genome-wide expression information from 33 brain regions classified into seven groups (*Khrameeva et al., 2020*) of four primate species, while the Mayo Data includes data from two brain regions (temporal cortex and cerebellum) of AD patients and controls. In brief, expression of all genes and TEs was quantified and normalized across samples of each dataset, and subsequently DE and correlations between KRAB-ZNFs and TEs were calculated using TEKRABber (*Figure 1A*, *Figure 1— figure supplement 2*).

## Dynamics of expression across species and brain regions: strong species differences especially of evolutionary young KRAB-ZNF genes and TEs

We obtained normalized expression values of KRAB-ZNF genes and TEs and assessed their variance across different species using t-SNE clustering (*Figure 2A*). The data labeled by species revealed distinct boundaries of variance, demonstrating clustering based on species. Specifically, humans and macaques formed their own clusters, while chimpanzees and bonobos were grouped in the same cluster, which agrees with phylogenetic distances among species. This finding indicated that clear differences in KRAB-ZNF genes and TE expression can be detected across species. For example, there were 12 upregulated and 42 downregulated KRAB-ZNF genes, along with 31 upregulated and 22 downregulated TEs in humans compared to chimpanzees in the primary and secondary cortices (*Figure 2B*; see *Supplementary file 1, table S2* for information on Brodmann areas included in the group 'primary and secondary cortices'). In contrast to the clear species differences, expression patterns of KRAB-ZNFs and TEs differed less across brain regions within the same species.

Certain TEs are primate-specific and have been extensively used in phylogenetic studies. For example, a subset of recently evolved Alu subfamilies is found only in Simiiformes (*Xing et al., 2007*; *Williams et al., 2010*). We next investigated whether expression patterns differ between Simiiformes-specific KRAB-ZNF genes and TEs (called evolutionary **young** from here on), and KRAB-ZNFs and TEs that have also orthologs outside Simiiformes (called evolutionary **old** from here on). To this end, we dated these genomic elements and classified them into the two groups based on their inferred evolutionary age (see Materials and methods). The old group consists of 234 KRAB-ZNF genes and 955 TEs that evolved prior to the emergence of Simiiformes (>44.2 million years ago [mya]), while the young group consists of 103 KRAB-ZNF genes and 309 TEs that evolved less than 44.2 mya (*Figure 2C*, *Figure 2—figure supplements 1 and 2*). We first found that the young KRAB-ZNFs and young TEs exhibited significantly lower expression levels across all brain regions, regardless of species (*Figure 2D*, *Figure 2—figure supplement 3*). Next, there were proportionally more young KRAB-ZNF genes and young TEs differentially expressed between humans and chimpanzees compared to old KRAB-ZNF genes and old TEs (*Figure 2E*). The same holds true when comparing humans to all NHPs, implying that young TEs and young KRAB-ZNFs are still more dynamically changed between different primates.

We further investigated if there are KRAB-ZNFs and TEs that are specifically changed in humans compared to the three NHPs. We detected 36 such KRAB-ZNF genes and 18 such TEs in the primary and secondary cortices. KRAB-ZNFs showed a trend toward more downregulation in humans, such as *ZNF337* and *ZNF394* (*Figure 2F*). Unlike this trend, some KRAB-ZNFs linked to cognitive disorders were human specifically upregulated, e.g., *ZNF778*, a candidate gene for autism spectrum disorder and cognitive impairment (*Willemsen et al., 2010*), and *ZNF267*, which is upregulated in the prefrontal cortex of AD patients (*Patel et al., 2021*). TEs tended to be more upregulated in humans compared to NHP. On the other hand, the LTR12B subfamily is one of the most downregulated TEs in humans. LTR12-related ERV subfamilies had been reported to be repressed by ZNF676 and ZNF728 in early human development (*Iouranova et al., 2022*).

## Changes in correlations between TEs and KRAB-ZNF genes: increased connectivity in the human brain co-expression network with an enrichment for evolutionary young correlations

To systematically analyze the putative functional relationships between TEs and KRAB-ZNF genes, we conducted pairwise Pearson's correlation analysis using normalized expression levels in seven clustered brain regions (*Supplementary file 1, table S2* and Figure 1 in *Khrameeva et al., 2020*). We first analyzed the human samples. There were 324 KRAB-ZNFs and 895 TEs (subfamily level) expressed in

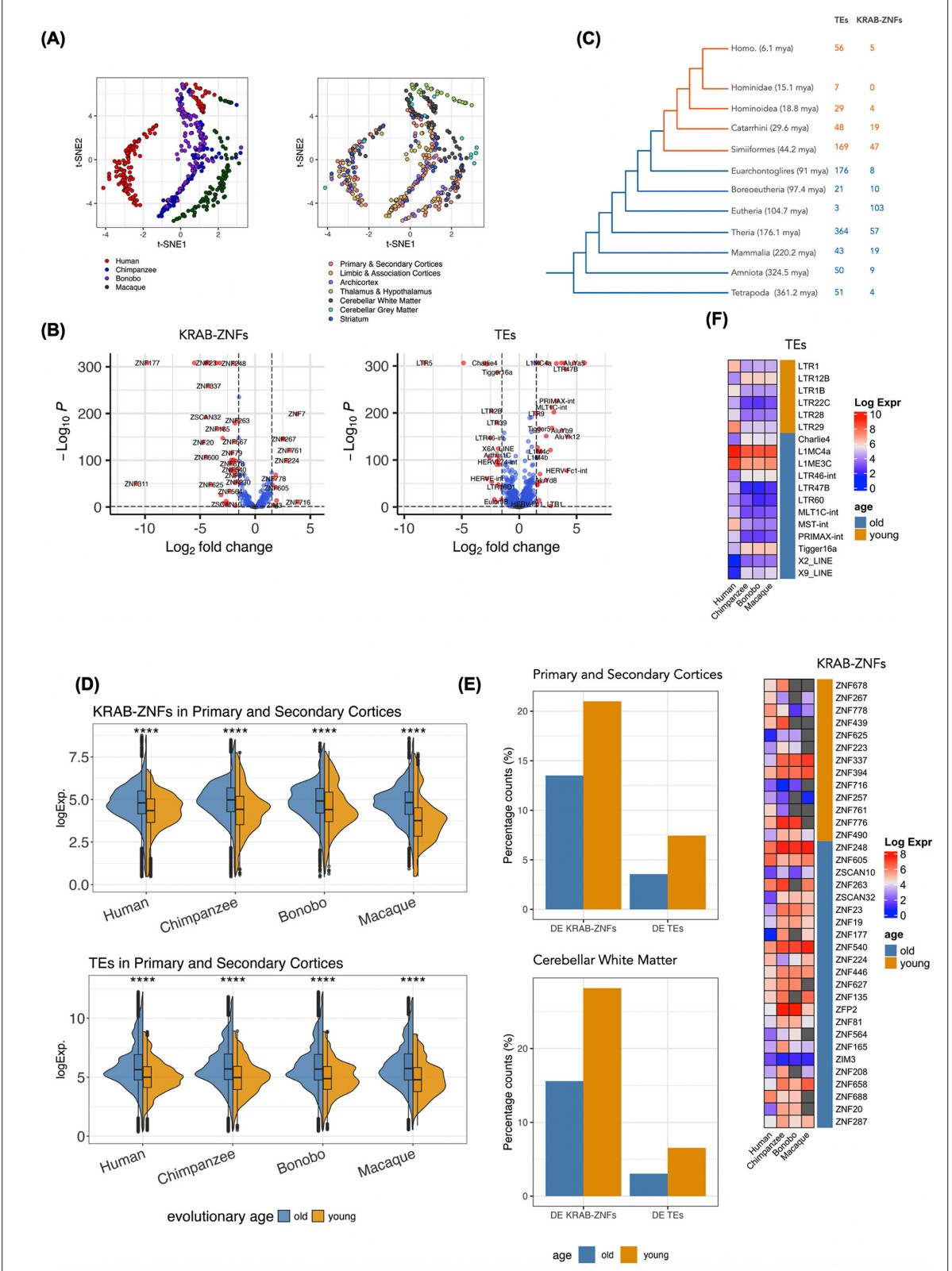

**Figure 2.** Expression of KRAB-ZNF genes and transposable elements (TEs) in Primate Brain Data. (**A**) t-SNE plots of the expression of KRAB-ZNF genes and TEs from all 422 samples, including human, chimpanzee, bonobos, and macaques labeled by species and different brain regions. (**B**) Differentially expressed KRAB-ZNF genes and TEs comparing human and chimpanzee in primary and secondary cortices. (**C**) Species tree with the inferred numbers of TEs and KRAB-ZNFs that have evolved per branch. (Note: There are 247 relatively old TEs and 52 KRAB-ZNFs that were difficult to place into a

*Figure 2 continued on next page*

*Figure 2 continued*

specific branch. Thus, they are not presented in this panel.) (**D**) Expression of KRAB-ZNF genes and TEs in primary and secondary cortices across species. Both KRAB-ZNF genes and TEs were grouped into two groups based on their inferred evolutionary age, old (>44.2 million years ago [mya]) and young (≤44.2 mya). Young KRAB-ZNFs and young TEs have lower expression levels (Wilcoxon rank sum test, p<0.05). Expressions of all brain regions can be found in *Figure 2—figure supplement 3*. (**E**) Percentage of differentially expressed KRAB-ZNF genes and TEs in humans compared to chimpanzees in primary and secondary cortices and cerebellar white matter. (**F**) Human-specific differentially expressed (DE) (i.e. human-specifically changed) KRAB-ZNF genes and TEs in primary and secondary cortices compared to nonhuman primates (NHPs). Gray indicates no expression information. The colors for age inferences in (**C**), (**D**), (**E**), and (**F**) are the same: blue for evolutionary old and orange for evolutionary young KRAB-ZNFs and TEs, respectively.

The online version of this article includes the following figure supplement(s) for figure 2:

**Figure supplement 1.** Distribution of KRAB-ZNFs evolutionary age inference.

**Figure supplement 2.** Distribution of transposable elements (TEs) evolutionary age inference.

**Figure supplement 3.** Expression of KRAB-ZNFs and transposable elements (TEs) among brain regions.

Primate Brain Data (copy number provided in *Supplementary file 1, table S4*). In humans, we found 100,987 positive and 26,810 negative significant correlations between TEs and KRAB-ZNFs in the primary and secondary cortices and 38,295 positive and 11,475 negative significant correlations in the limbic and association cortices (adjusted p-value<0.01), while the other clusters have fewer or no correlations detected (*Supplementary file 1, table S5*). We will thus mainly focus on the primary and secondary cortices and the limbic and association cortices for our subsequent analyses.

The numbers of correlations between TEs and KRAB-ZNFs are significantly more than expected by chance, as gauged by repeating the correlation analysis with randomly picked genes and TEs (p<0.001; see Materials and methods; *Figure 3A*, *Figure 3—figure supplement 1*), indicating that putative functional relationships between TEs and KRAB-ZNFs can be detected in the data. The high number of positive correlations might be surprising, given that KRAB-ZNFs are considered to repress TEs. However, the dataset contains in general more positive correlations, even when choosing random genes, and KRAB-ZNFs still have more negative correlations to TEs than random genes. It is also plausible that some relationships between older KRAB-ZNFs and no longer harmful TEs are positive, e.g., due to embedding into functional pathways, when a repression is no longer needed.

To remove weak correlations, we set a threshold with absolute correlation coefficients greater than 0.4 and adjusted p-value below 0.01 (Benjamini-Hochberg correction) (*Figure 3B*, *Figure 3—figure supplements 1 and 2*). In addition, we sought independent experimental confirmation of our detected putative relationships by overlapping the correlations with experimental ChIP-exo data obtained from KRAB-ZNF proteins in human embryonic stem cells (*Imbeault et al., 2017*). Note that although ChIP-exo was performed in different cell types than the brain samples we analyzed, the overlap was significant by calculating their Jaccard similarity (p<0.001, *Figure 3C*). This result supports that our correlations have significantly captured the interplay between TEs and KRAB-ZNFs. Focusing on the ChIP-exo-confirmed correlations, we obtained 869 correlations in the primary and secondary cortices and 399 correlations in limbic and association cortices. Of those, 201 positive correlations and 166 negative correlations overlapped between these two brain region groups (*Figure 3D*).

To better understand the complexity of the interactions between TEs and KRAB-ZNFs, we represented the one-to-one pairwise correlations of TE and KRAB-ZNF (denoted as **TE:KRAB-ZNF** in the following content) in a bipartite network. In this bipartite network, nodes belong to one of two classes, TEs or KRAB-ZNFs, and only links between nodes of different classes are presented. Our bipartite network of human primary and secondary cortices reveals 354 nodes connected by 869 links (*Figure 3E*), and the human limbic and association cortices have 247 nodes with 399 links (*Figure 3—figure supplement 3*). To identify the hubs in the network, we first calculated their normalized degree distributions, considering the unequal numbers of TEs and KRAB-ZNFs. Our analysis revealed that KRAB-ZNF nodes connect to more TEs than TE nodes connect to KRAB-ZNFs, indicating that KRAB-ZNF nodes are more likely to act as hubs with higher connectivity (*Figure 3F*). For investigating the structure of the network, we calculated the bipartite modularities based on the correlation coefficients. Our findings revealed that the network of the primary and secondary cortices is clustered into 5 modules (*Figure 3E*), while the network of the limbic and association cortices clusters into 13 modules (*Figure 3—figure supplement 3*).

We divided TE:KRAB-ZNF into evolutionary young and old, calling a correlation as **young** when one of the nodes is evolutionary young and **old** when both nodes are evolutionary old (*Figure 3—figure*

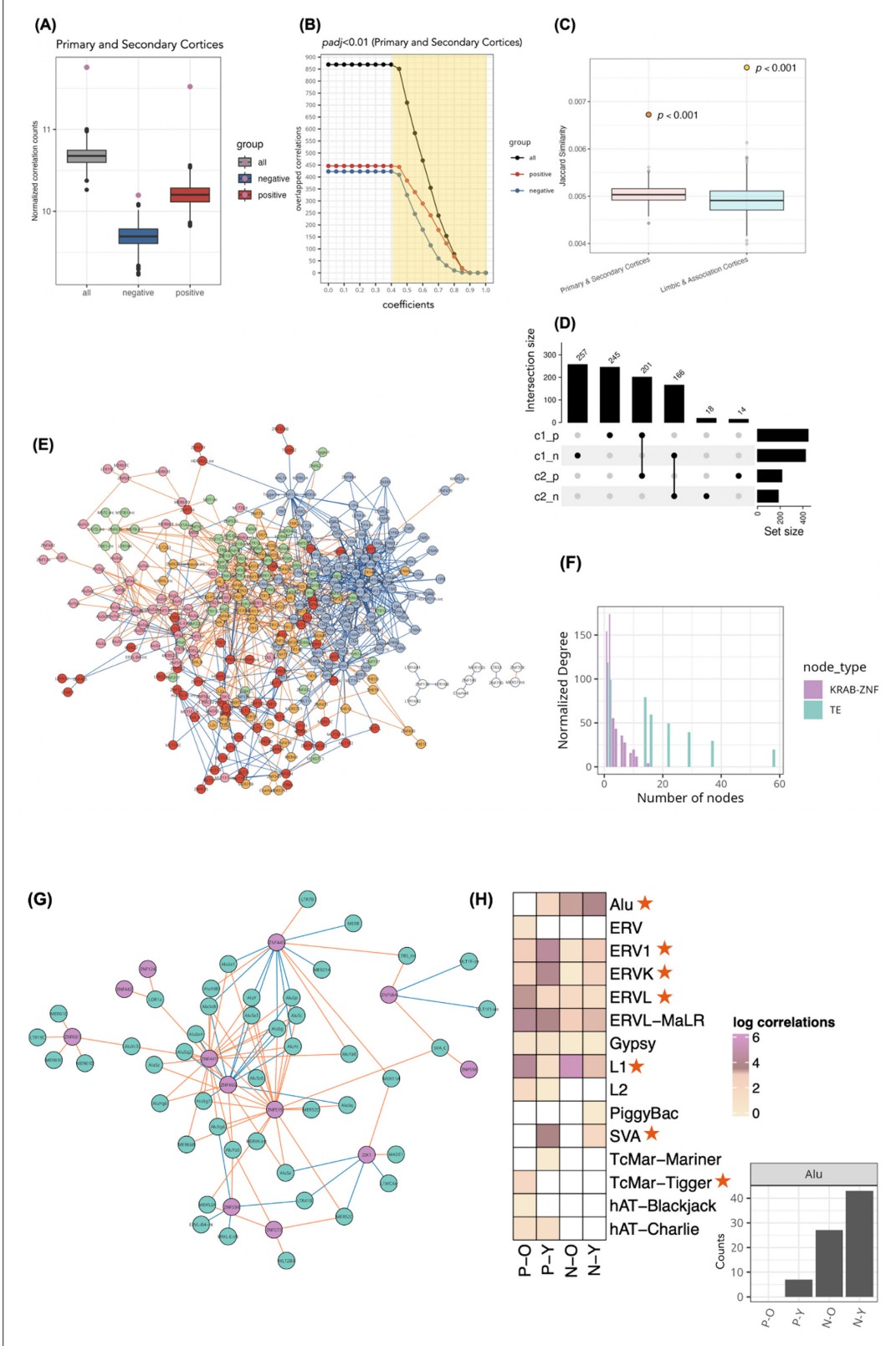

**Figure 3.** TE:KRAB-ZNF in human primary and secondary cortices. (**A**) and (**B**) demonstrate the workflow for checking significant TE:KRAB-ZNF using the primary and secondary cortices as an example. (**A**) We used randomly selected gene sets and KRAB-ZNFs to calculate correlations with transposable elements (TEs). The violet dots indicate the correlation counts of TE:KRAB-ZNF based on comparing all correlations, positive correlations, and

*Figure 3 continued on next page*

*Figure 3 continued*

negative correlations. They are significantly higher than for random gene sets (boxplots below, 1000 iterations, p<0.001). (**B**) Overlaps between TE:KRAB-ZNF (y-axis) and the KRAB-ZNF protein ChIP-exo data (*Imbeault et al., 2017*) (x-axis). Note that we use absolute coefficient values for negative correlations. Correlations under the yellow area are selected (**C**) Jaccard similarity, demonstrating that the correlations between TEs and KRAB-ZNFs overlapped significantly more with ChIP-exo data than randomly selected TEs and KRAB-ZNFs (p<0.001). The points indicate the overlap with actual correlations and ChIP-exo data. The boxplots indicate the overlap between randomly selected TE and KRAB-ZNF pairs with ChIP-exo data. (**D**) Subsets of the number of positive and negative TE:KRAB-ZNF in the primary and secondary cortices (c1_p and c1_n) and the limbic and association cortices (c2_p and c2_n). (**E**) TE:KRAB-ZNF network in the primary and secondary cortices with five modules. Nodes are colored in five colors representing the five modules, and nodes in white do not belong to any module. Young links are in orange and old links are in blue. (**F**) Distribution of the normalized degree counts in TE and KRAB-ZNF nodes in TE:KRAB-ZNF network. (**G**) This is the subnetwork colored in pink from (**E**), showing that this module mainly consists of Alu subfamilies. (**H**) The log count of correlations classified by TEs and the categories of links, including positive-old (P-O), positive-young (P-Y), negative-old (N-O), and negative young (N-Y). Red stars indicate that the class distribution of the TEs is significantly different (Chi-squared test, p<0.001). The right-hand side barplot shows the exact count of Alu subfamilies from the first row in the heatmap.

The online version of this article includes the following figure supplement(s) for figure 3:

**Figure supplement 1.** Distribution of correlation in limbic and association cortices.

**Figure supplement 2.** Example of correlations (TE:KRAB-ZNF).

**Figure supplement 3.** TE:KRAB-ZNF network in human limbic and association cortices.

---

*supplement 2*). Taking also the correlation coefficients into account, all links can be classified into four categories: positive-old (P-O), positive-young (P-Y), negative-old (N-O), and negative-young (N-Y). We found that the links are nonrandomly distributed in the network. In particular, Alu, ERV1, ERVK, ERVL, L1, TcMar-Tigger, and SVA have significantly different interactions with KRAB-ZNF than the other TE subfamilies (Chi-squared test, p<0.001). For example, in a module with Alu subfamilies (*Figure 3G*), we observed that most of the correlations specifically belong to N-Y, while SVA had many P-Y correlations with KRAB-ZNFs (*Figure 3H*).

Next, we compared the TE:KRAB-ZNF between species and only considered the 178 KRAB-ZNFs and 836 TEs, which were expressed in all four species. Since the original study (*Khrameeva et al., 2020*) included four human individuals but only three individuals per NHP species, we performed a leave-one-out analysis of the human samples. For a fair comparison across species, we required that a correlation between KRAB-ZNFs and TEs in humans needed to be significantly detected in all human leave-one-out combinations (*Figure 4A*, *Supplementary file 1, table S6*). We then repeated our test of whether KRAB-ZNFs are more likely to correlate with TEs compared to randomly selected genes and found that human KRAB-ZNF still has a significantly higher number of correlations with TEs. In contrast, in NHPs, KRAB-ZNFs did not have more correlations to TEs than randomly selected genes (*Figure 4—figure supplement 1*).

We subsequently determined human-specific and conserved TE:KRAB-ZNF interactions by assessing whether an interaction seen in humans also existed in any NHP. Remarkably, the results we obtained show that humans have a higher number of correlations and connectivity compared to NHPs (*Figure 4B*). This finding is not confounded by a lack of gene/TE annotations nor sample size, given that we only included KRAB-ZNF genes and TEs expressed in all four species, and that we used the same number of individuals per species; being even more conservative by requiring that all four permutations of human triples show a significant correlation.

Interestingly, we found that some correlations had opposite signs between species. For example, 276 TE:KRAB-ZNF with positive correlations in humans were negatively correlated in bonobos (the seventh column in *Figure 4B*). Although not significant, out of the 276 TE:KRAB-ZNF, 104 were also negatively correlated in chimpanzees and rhesus macaques and might represent human-specific changes in the sign of the correlation. Constructing a network of those 276 TE:KRAB-ZNF, we detected that *ZNF112* and *ZNF528* are two hubs connected to TEs with mostly old links and young links, respectively (*Figure 4C*). We asked whether sequence differences exist between the orthologous zinc finger proteins that might explain this putative functional change. For ZNF112, there were 21 zinc finger domains. However, among 14 amino acid differences, none of them affected a position within the zinc

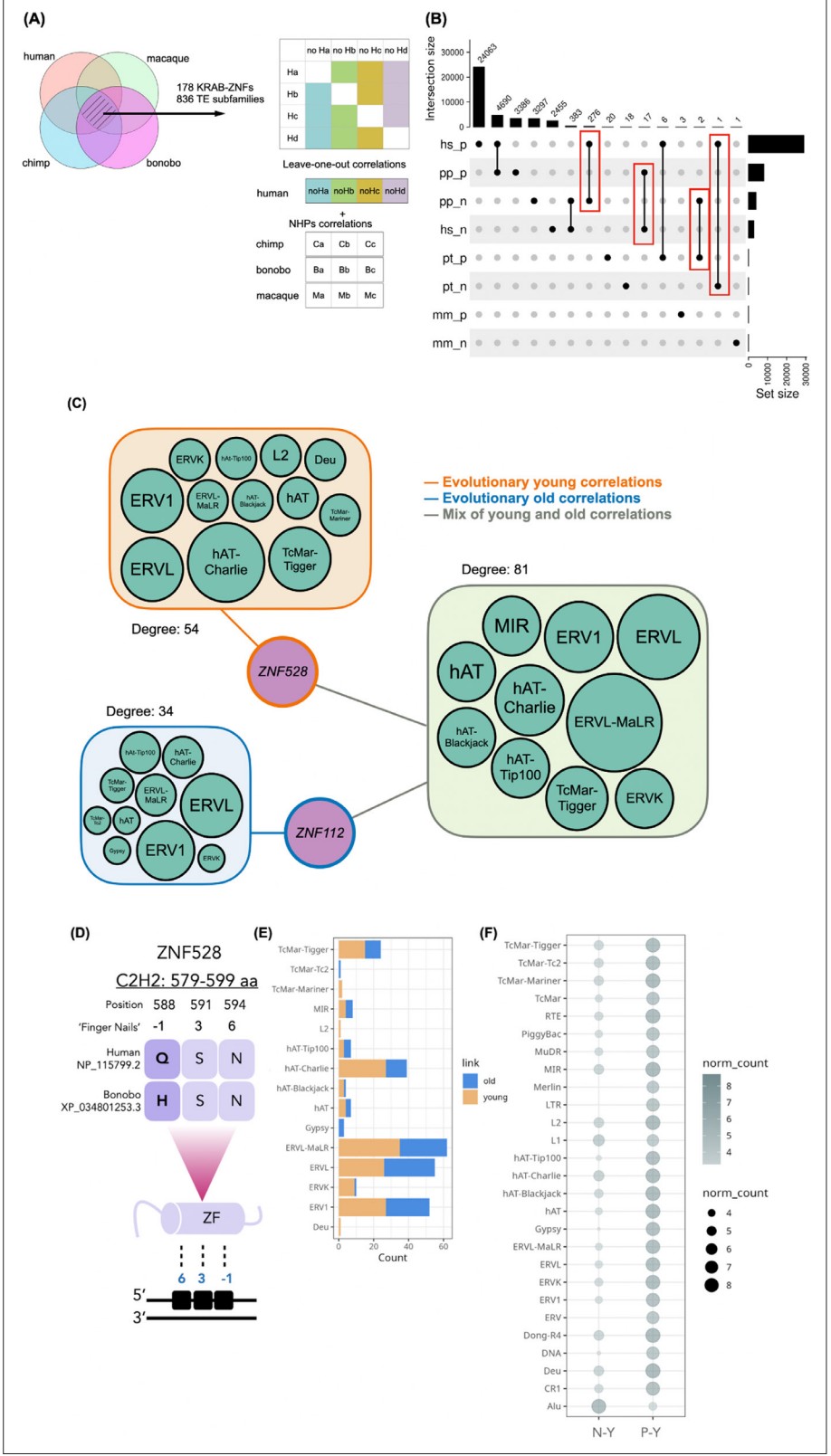

**Figure 4.** Comparison of TE:KRAB-ZNF in human and nonhuman primates (NHPs). (**A**) Workflow for selecting TE:KRAB-ZNF comparing between species. First, there were 178 transposable elements (TEs) and 836 KRAB-ZNFs detected in all four species. Second, the leave-one-out test in the human sample was performed for a fair comparison since humans had four repeats and NHPs only had three (adjusted p<0.01 and absolute

*Figure 4 continued on next page*

*Figure 4 continued*

coefficient >0.4) (**B**) Number of positive and negative correlations in human and NHPs. Red brackets indicated the change of correlation between two species. For example, there are 276 human positive correlations which are negatively correlated in bonobos (suffix n: negative, p: positive; hs: *Homo sapiens*, pt: *Pan troglodytes*, pp: *Pan paniscus*, mm: *Macaca mulatta*). (**C**) Network of 276 TE:KRAB-ZNF that were all positively correlated in humans but negatively correlated in bonobos. This network demonstrates two hubs, *ZNF528* and *ZNF112*, connecting to multiple TE subfamilies. Node size of TEs refers to the relative abundance of connections to the hubs. Details of this network can be found in *Figure 4—figure supplement 2*. (**D**) ZNF528 protein sequence difference in a zinc finger domain (ZF), where humans have glutamine (**Q**) while bonobos have histidine (**H**) at the –1 finger position. The lower part of the illustration indicates that the zinc finger domain binds to the DNA sequence using the –1, 3, and 6 finger positions. (**E**) Number of different TE subfamily nodes which form evolutionary old and young correlations in (**C**) network comparing humans to bonobos. (**F**) Distribution counts of human-specific correlations categorized based on TE subfamilies showing only young links. N-Y: negative and young correlations; P-Y: positive and young correlations.

The online version of this article includes the following figure supplement(s) for figure 4:

**Figure supplement 1.** Comparing KRAB-ZNFs and randomly selected genes in Primate Brain Data.

**Figure supplement 2.** 276 opposite TE:KRAB-ZNF regulatory network comparing humans to bonobos.

---

finger domains (–1, 3, and 6) that directly contacts the DNA. Between the orthologs of human and bonobo ZNF528, we discovered an amino acid difference in one position that directly contacts nucleotides of the DNA (–1 position of the 15th zinc finger domain) (*Figure 4D*). While in bonobo ZNF528, there is a histidine at this position, it is a glutamine in human ZNF528. For other KRAB-ZNF proteins, it has been demonstrated that replacing DNA-contacting amino acids with alanine or glutamine reduces their repressor potency (*Nunez et al., 2011*). Therefore, we speculate that variations in the zinc finger domain of ZNF528, specifically at position 588, may explain why human ZNF528 is not as negatively correlated with TEs as bonobo ZNF528 (*Figure 4D*). Interestingly, this changed position represents a very rare human polymorphism (rs373201614), which seems to be under positive selection in humans, including Denisovans and Neanderthals (*Harrison et al., 2024*).

The distribution of young and old links was not random in the 276 TE:KRAB-ZNF bipartite network (*Figure 4C*). For instance, TcMar-Tc2 and Gypsy had only old interaction with KRAB-ZNFs, while most of the ERVK correlations were classified as young (*Figure 4E*).

Last, we checked the species-specific correlations based on TE subfamilies. For example, there are 24,063 positive and 2455 negative correlations in humans that were not detected in NHPs (the first column and the fifth column in *Figure 4B*). Interestingly, these human-specific correlations were all evolutionary young links, and many of them represented negative correlations involving TEs of the Alu subfamilies (*Figure 4F*).

## Alterations in TE and KRAB-ZNF expression in brains of Alzheimer's patients reflect brain region differences

Several KRAB-ZNFs and TEs with human-specific expression patterns or correlations have been associated with AD. For example, *ZNF267*, which was upregulated in the human cortex compared to NHPs (*Figure 2F*), is a clear transcriptomic signature for the diagnosis of AD (*Fehlbaum-Beurdeley et al., 2012*), and the expression of AluYa5 subfamily leads to genetic dysregulation in AD (*Kim et al., 2016*). We thus hypothesized that the regulatory network of KRAB-ZNFs and TEs might be severely altered in the brains of AD patients. We utilized the **Mayo Data** to test this hypothesis (*Table 2*).

To investigate if there were differences in the expression patterns of TEs and KRAB-ZNFs comparing different brain regions and disease status, we conducted t-SNE analysis using their expression levels from the temporal cortex and cerebellum of human control individuals and AD patients. Results showed that variances were primarily influenced by brain regions rather than AD status (*Figure 5A*). Similar to our previous results with the Primate Brain Data (*Figure 2D*), we found that evolutionary young KRAB-ZNF genes and young TEs were expressed at lower levels than their older counterparts (*Figure 5B*). Comparing expression levels between AD and control samples, we obtained for the temporal cortex 6 KRAB-ZNF genes and 4 TEs that were upregulated in AD and 6 TEs that were downregulated in AD (*Figure 5C*). In the cerebellum, there were 6 upregulated and 1 downregulated TEs, and 4 downregulated and 10 upregulated KRAB-ZNF genes (*Figure 5D*).

## Human-specific correlations limited to the healthy human temporal cortex: 21 TE:KRAB-ZNFs not detected in AD

To select significant correlations (TE:KRAB-ZNF) between control and AD samples in the temporal cortex and cerebellum, we employed the same filtering criteria as described in the Primate Brain Data analysis (*Figure 3A–C*), requiring an adjusted p-value less than 0.01, absolute correlation coefficients higher than 0.4, and TE:KRAB-ZNF pairs detected in ChIP-exo data (*Imbeault et al., 2017*). The overlaps of TE:KRAB-ZNF pairs are depicted in *Figure 6A*, demonstrating a higher number of correlations in the control group. Next, we selected 21 TE:KRAB-ZNF, which are detected from the temporal cortex both in human Primate Brain Data and the healthy controls in Mayo Data, but not detected in any NHPs in Primate Brain Data. These correlations represented a subset of TE:KRAB-ZNF, which were specific to healthy adult human brain samples but not detected in AD progression (*Figure 6B*). Among these 21 TE:KRAB-ZNF, there are 14 evolutionary young and 7 evolutionary old interactions, and we found that Alu subfamilies accounted for 11 of these 21 interactions (*Figure 6C*). We further investigated why these TE:KRAB-ZNF were not detected in AD samples and found that most of the correlations were not significant based on our defined threshold. For instance, AluYc:*ZNF182* exhibited a positive correlation in both control and AD samples in the temporal cortex. However, according to our selection criteria, this TE:KRAB-ZNF pair was not deemed significant in the AD sample (FDR = 0.34) (*Figure 6D*). The sole exception, showing opposite direction of correlation between groups, was L1MA6:*ZNF211*, which displayed a negative correlation in the control group but had a nonsignificant very weak positive correlation in the AD group (*Figure 6D*). This indicates weakening and loss of some correlations between TEs and KRAB-ZNFs in AD.

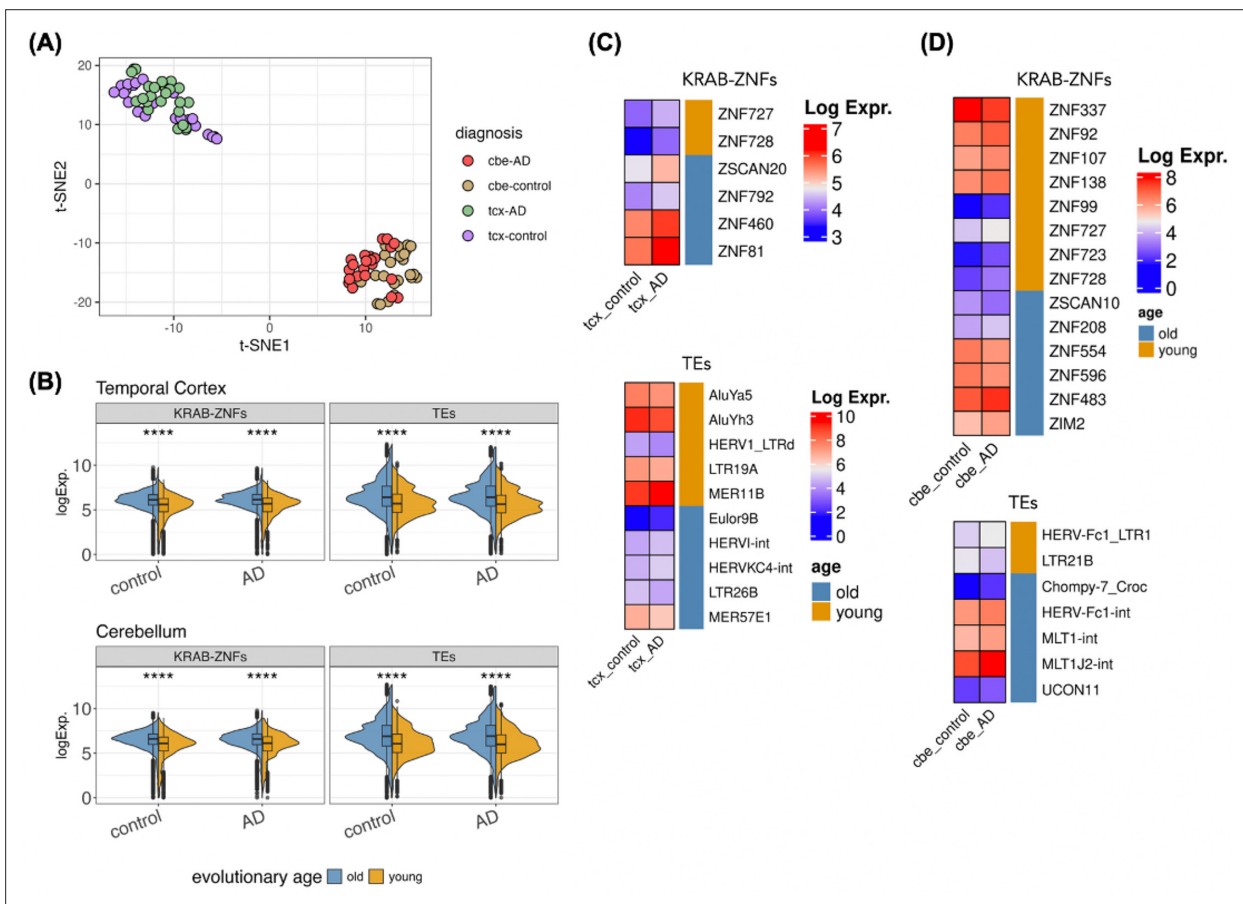

**Figure 5.** Expression of transposable elements (TEs) and KRAB-ZNF genes in Mayo Data. (A) Variations in the expression of KRAB-ZNF genes and TEs using t-SNE analysis. cbe: cerebellum; tcx: temporal cortex. (B) Distributions of the expression of evolutionary old and young KRAB-ZNF genes and TEs. (C) and (D) Differentially expressed KRAB-ZNF genes and TEs (absolute log2FoldChange>0.5, p<0.05) in temporal cortex (tcx) and cerebellum (cbe). The expression of KRAB-ZNF genes and TEs in the cerebellum shared the same log expression scale.

**Table 2.** Datasets.

| Dataset | Categories (biological replicates) | Total number of samples |
|---|---|---|
| Primate Brain Data (GSE127898) | Human (4) | 132 |
| | Chimpanzee (3) | 96 |
| | Macaque (3) | 96 |
| | Bonobo (3) | 98 |
| Mayo Data (syn5550404) | Control-temporal cortex | 23 |
| | AD temporal cortex | 24 |
| | Control-cerebellum | 23 |
| | AD cerebellum | 22 |

Subsequently, we conducted a bipartite network analysis using the 21 TE:KRAB-ZNF pairs specific to the healthy human samples in the temporal cortex. Integrating evolutionary age inference, we identified a module of young Alu subfamilies in the temporal cortex characterized by negative correlations to KRAB-ZNFs (*Figure 6E–F*). Interestingly, the negative young TE:KRAB-ZNF pairs in this Alu module are not significantly detected in AD brains, suggesting that the regulation of the involved TEs is lost or at least impaired in the disease condition.

## Discussion

In this study, we conducted systematic examinations of TEs and KRAB-ZNFs expression and correlation patterns in the brain and compared them between humans and NHPs, as well as between healthy human samples and AD. We demonstrated high divergence in the expression of TEs and KRAB-ZNF genes between species, suggesting a prominent evolutionary signature in the regulation of TEs and KRAB-ZNF genes in primates (*Figure 2A*). We also found that evolutionary younger members of both TEs and KRAB-ZNF genes exhibit significantly lower expression levels compared to older members (*Figure 2D*), proposing that newly emerged genetic elements might be subjected to more stringent regulatory mechanisms, potentially avoiding disruption at the organismal level. To represent the complexity of interactions between TEs and KRAB-ZNFs, we derived bipartite networks. We showed that humans had higher connectivity in these networks compared to NHPs (*Figure 4B*). This observation is in line with previous findings that humans also have higher connectivity in the transcription factor networks of the brain compared to NHPs (*Nowick et al., 2009*; *Bakken et al., 2016*; *Berto et al., 2018*). A substantial proportion of young TE:KRAB-ZNF seems to be human-specific, pointing to recent additions to the TE:KRAB-ZNF network in the human lineage and presumably an increase in the complexity of gene regulation (*Figure 4F*). The network of the 21 TE:KRAB-ZNF that were only discovered in the healthy human temporal cortex but not in other NHPs (*Figure 6E and F*) indicates potential alterations or loss of control of some TE expression in AD brains. These collective findings suggest an important role of the TE:KRAB-ZNF network in shaping the evolution and in maintaining the functionality of the healthy human brain, and particularly highlight the co-expression involving evolutionary young TEs and young KRAB-ZNFs.

Given the generally acknowledged role of KRAB-ZNF proteins as repressors on TE expressions, we expected to observe many negative correlations between KRAB-ZNFs and TEs. Indeed, many evolutionary young and human-specific correlations involving especially TEs of the Alu subfamilies are negative. However, we also observed numerous positive correlations between KRAB-ZNFs and TEs. Keeping in mind that correlations do not indicate direction of the causation and can also be caused by indirect relationships and other factors, several plausible explanations could be offered for these positive interactions. First, some TEs can function as *cis*-acting regulatory elements capable of influencing the expression of nearby genes, which have been mentioned to be tissue-specifically co-opted TEs (*Sentmanat and Elgin, 2012*; *Wolf et al., 2020*; *Coronado-Zamora and González, 2023*). Second, it is conceivable that the expression levels of KRAB-ZNF genes themselves are influenced by the presence and activity of TEs within the genome (*Pontis et al., 2019*; *Senft and Macfarlan, 2021*). Another perspective is that primate-specific KRAB-ZNFs can bind to gene promoters, regulating gene

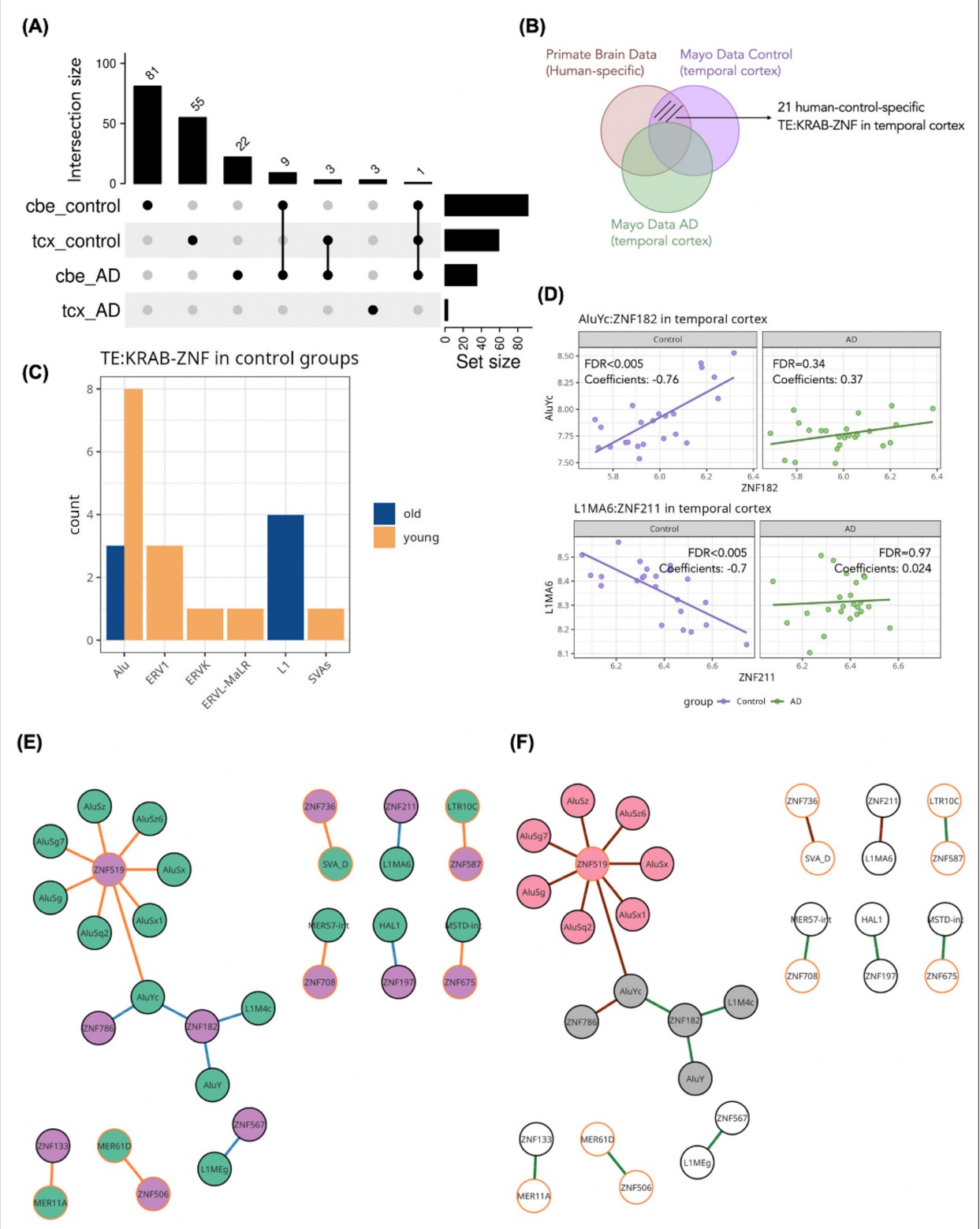

**Figure 6.** TE:KRAB-ZNF analysis in Mayo Data. (**A**) Overlaps of TE:KRAB-ZNF between control and Alzheimer's disease (AD) condition in temporal cortex and cerebellum (denoted as cbe_control, tcx_control, cbe_AD, and tcx_AD). (**B**) 21 human-control-specific TE:KRAB-ZNFs were selected from the intersection of human-specific TE:KRAB-ZNFs from Primate Brain Data (not detected in the other nonhuman primates [NHPs]) and control-specific TE:KRAB-ZNFs from Mayo Data in temporal cortex (not detected in AD samples). (**C**) Distribution of transposable element (TE) families counts among the 21 control-specific TE:KRAB-ZNF in the temporal cortex. (**D**) Comparison of the expression and correlation results of AluYc:*ZNF182* and

*Figure 6 continued on next page*

*Figure 6 continued*

L1MA6:*ZNF211* in the temporal cortex. (**E**) and (**F**) show the bipartite network of 21 TE:KRAB-ZNF in the temporal cortex. (**E**) Coloring TE nodes in green and KRAB-ZNF nodes in violet. Evolutionary young links are in orange, and evolutionary old links are in blue. Orange border specified that this TE or KRAB-ZNF evolved recently. (**F**) There are two modules in the network colored in pink and gray based on their bipartite modularity. Brown links indicate negative correlations, and green links are positive correlations.

expression in the primate brain without necessarily targeting TEs (*Farmiloe et al., 2020*); observed correlations could then stem from indirect regulation. We should note, however, that 19.43% (446/2,296) of positively correlated pairs overlap significantly with the ChIP-exo results, i.e., exist in the presence of a KRAB-ZNF protein binding to the TE (*Figure 3B*). Furthermore, we discovered that all the human-specific correlations are evolutionary young correlations. Notably, different TE subfamilies interact with KRAB-ZNFs differently, e.g., Alu subfamilies are the only TEs that have more negative correlations than positive ones, and LTR subfamilies have only positive correlations (*Figure 4F*). Therefore, we encourage future research focusing on primate evolution to take these factors into account.

TEs have also been observed to be dysregulated in certain diseases. Here, we observed that the variances in expression levels of TEs and KRAB-ZNFs were more distinct across various brain regions than between control and AD samples (*Figure 5A*). This finding points out the difficulties in identifying differences between control and AD samples solely through the consideration of DE. This is also illustrated by the detection of only 6 and 10 differentially expressed KRAB-ZNFs and TEs, respectively (*Figure 5C*). Upon combining expression data with correlation results and an evolutionary perspective, we determined 21 human-specific TE:KRAB-ZNFs that were not detectable in AD brains (*Figure 6B, E, and F*), suggesting considerable alterations of connections in the context of a human-specific disease. By comparing healthy and disease samples, we discovered evolutionary young Alu subfamilies that are downregulated in AD patients, such as AluYh9 and AluSp subfamilies, which have been found to be negatively associated with Tau pathologic burden in the human dorsolateral prefrontal cortex (*Guo et al., 2018*). The AluY and AluSx subfamilies derived Alu-mediated in-frame deletions of the exons 9 and 10 in mutated forms of *PSNE1*, which was associated with early-onset AD (*Le Guennec et al., 2017*). Alu subfamilies have further been shown to regulate the expression of *ACE* in neurons from Alu insertions (*Wu et al., 2013*) and to play a role in A-to-I RNA editing during neurogenesis in the central nervous system and mitochondrial homeostasis (*Larsen et al., 2018*). Thus, we suggest the dysregulation of modules of Alu subfamilies with their connected KRAB-ZNFs in the regulatory network (*Figure 6F*) plays a role in AD progression.

With respect to the arms-race hypothesis (*Jacobs et al., 2014*), we observed a higher number of negative correlations between young TEs and young KRAB-ZNF genes in humans compared to NHPs. This suggests increased repression from young KRAB-ZNFs on young TEs in the human brain (*Figure 4B and F*). Still, our findings only weakly support the arms-race hypothesis. First, we noted that young TEs and KRAB-ZNFs exhibit lower expression levels than their old counterparts (*Figures 2D and 5B*). This pattern may not align with the expectation that young TEs have recently escaped repression, which would likely result in higher expression of the young TE or the young KRAB-ZNF to keep the TE in check. However, it remains challenging to discern at which stage of a potential arms race the TE:KRAB-ZNF pair presently is, i.e., whether the TE is on the rise with its expression or already repressed by the KRAB-ZNF. In addition, older TEs might be allowed to be expressed more highly as they are not harmful anymore. Not in line with the arms-race model is further evidence that some young TEs were also negatively correlated with old KRAB-ZNF genes, leading to weak assortativity regarding age inference. We acknowledge, however, that this is not a contradiction, since old KRAB-ZNF genes might be repurposed to repress young TEs, which could be more 'cost-effective' than evolving a new KRAB-ZNF gene. Potentially, restricting the analysis to KRAB-ZNFs and TEs younger than 25 mya would be more suitable for investigating the arms-race hypothesis in humans. During this period, we might expect a more direct arms-race scenario, as the evolutionary pressures between rapidly evolving TEs and their regulatory mechanisms should be more pronounced. However, it is important to note that such an analysis would have been limited by the detectable correlations of only 3 out of 9 KRAB-ZNF genes and 9 out of 92 TE subfamilies in our dataset (*Supplementary file 1, table S7*).

In response to the need for systematically comparing expression profiles of orthologous genes and TEs across diverse species or conditions, we developed TEKRABber, an R Bioconductor package that equips users to effortlessly adapt its comprehensive pipeline for their analyses. TEKRABber efficiently

processes mapped raw counts from various TE quantification tools and normalizes transcriptomic data across species by extending the usage of the scaling-based normalization method (*Zhou et al., 2019*). To optimize computational efficiency and parallel computing during the calculations of pairwise correlations, we harnessed the capabilities of RCpp (*Eddelbuettel and François, 2011*) and doParallel (*Corporation and Weston, 2011*). This approach allows for effective scaling, accommodating larger datasets and leveraging additional computational resources as needed. Taking these advantages, users can adapt the TEKRABber's pipeline to analyze selected orthologous genes and TEs from their expression data within an acceptable time frame. Taking one of the examples in this study, calculating pairwise correlations of 178 orthologs with 836 TEs (neglecting sample number) can be executed in approximately 5 min on standard hardware (Intel Core i7 2.6 GHz, 16 GB memory). Future studies could use TEKRABber, for instance, to investigate correlations between TEs and all genes, to identify candidates of genes whose expression is influenced by certain TEs. Another option would be to explore which factors might repress TE expression in invertebrates, which lack KRAB-ZNF proteins.

Nonetheless, it is crucial to recognize that our present approach is constrained by certain limitations. These constraints primarily stem from variations in the lengths and positions of the same TE across individual genomes of the same species, leaving some room for improving the normalization of TE expression levels. We only used the available reference genomes for our analysis; however, it becomes increasingly clear that substantial individual differences in the presence of TEs exist, which are not covered by a single reference genome of a species. Currently, there are emerging methods designed to detect TE expression, including tools for advancements in sequencing resolution like long-read sequencing (*Marx, 2023*), de novo TE annotations (*Storer et al., 2022*; *Orozco-Arias et al., 2024*), and locus-specific expression detection in single-cell RNA-seq analysis (*Rodríguez-Quiroz and Valdebenito-Maturana, 2022*). These developments hold promise for achieving a more precise depiction of the variations between samples and the reference, ultimately enhancing our understanding of TE-associated expression patterns. Hence, we acknowledge the significant potential of integrating improved TE orthologous information into our method.

## Conclusion

In summary, our findings underscore the intricate network of interactions between TEs and KRAB-ZNFs in both human evolution and neurodegenerative disease. To achieve a comprehensive understanding of TEs and KRAB-ZNFs functions, it is not enough to only examine expression levels, but network analysis as facilitated by TEKRABber needs to be leveraged. We found that the human brain exhibits a notably denser TE:KRAB-ZNF network compared to NHPs, particularly for more recently evolved TEs and KRAB-ZNFs. The healthy human brain TE:KRAB-ZNF network contains a distinct module composed exclusively of Alu subfamilies, which is an evolutionary novelty not observed in AD brains. These insights highlight the nuanced dynamics of TE:KRAB-ZNF interactions and their relevance in both evolutionary and disease contexts. We emphasize that TEs can have a role in species evolution and provide a tool, TEKRABber, to further investigate this across a larger number of taxa.

## Materials and methods

### Primate Brain Data

For comparing humans with NHPs, we used published RNA-seq data (GSE127898) including 422 brain samples from biological replicates that consisted of 4 humans, 3 chimpanzees, 3 bonobos, and 3 macaques (*Table 2*). Samples from each individual were from 33 different brain regions, and total RNA was sequenced on the Illumina HiSeq 4000 system with a 150 bp paired-end sequencing protocol. More details about samples and the preparation steps can be found in *Supplementary file 1, table S1* and *Khrameeva et al., 2020*. RNA-seq FASTQ data on Gene Expression Omnibus (*Barrett et al., 2012*) were retrieved using NCBI SRA Toolkit v3.0.3.

### Mayo Data

The Mayo RNA-seq study (*Allen et al., 2016*) was utilized to compare human control and AD samples. Total RNA was sequenced from both control and AD samples collected from the temporal cortex and cerebellum. The preprocessed FASTQ files, which include only control and AD samples in the temporal

cortex and cerebellum, were downloaded via Synapse consortium studies (accession: syn5550404). The number of samples can be found in *Table 2* and more details, including biological sex, age, and Braak staging, are in *Supplementary file 1, table S3*.

## Transcriptome analysis

Adapters and low-quality reads were removed from the FASTQ files using fastp v0.12.4 (*Chen et al., 2018*) with default parameters. The selected FASTQ files from both datasets were then mapped to their respective references downloaded from UCSC Table Browser, including hg38, panTro6, panPan3, and rheMac10 (*Karolchik et al., 2004*), using STAR v2.7.10b (*Dobin et al., 2013*) with the parameters '--outFilterMultimapNmax 100 --winAnchorMultimapNmax 100'. These parameters were chosen to increase the likelihood of capturing TE mapping by allowing for multiple alignments of reads and more loci anchors for mapping. The resulting BAM files were used to quantify counts of TEs and genes using TEtranscripts v2.2.3 (*Jin et al., 2015*) with the '--sortByPos' parameter to determine the expression levels of genes and TEs. Gene and TE indices were created using UCSC gene annotations and the RepeatMasker track, enabling reads to match with these intervals. If a read overlapped with both a gene exon and a TE, we determined whether it had a unique alignment or multiple locations in the genome. If an annotation existed, the uniquely aligned read was assigned to the gene. Otherwise, it was assigned to the TE. For reads with ambiguous mappings, they were evenly weighted across TE or gene annotations using the expectation maximization algorithm (*Dempster et al., 1977*). Differentially expressed genes and TEs were quantified using DESeq2 v1.4.4 (*Love et al., 2014*), utilizing an absolute log2FoldChange threshold greater than 1.5 (adjusted p<0.05). However, the Mayo Data adopted an absolute log2FoldChange threshold greater than 0.5 due to a lower number of detected differentially expressed genes and TEs. Heatmaps and upset plots were visualized using ComplexHeatmap v2.2.0 (*Gu et al., 2016*; *Gu, 2022*).

## Inferences about the evolutionary age of KRAB-ZNFs and TEs

A total of 337 KRAB-ZNF genes were identified using the comprehensive KRAB-ZNF catalog (*Huntley et al., 2006*). The evolutionary age of these genes was inferred through annotations provided by GenTree (*Shao et al., 2019*), which employed a synteny-based pipeline to date primate-specific protein-coding genes and depicted their origins using a branch view. For KRAB-ZNF genes lacking direct dating annotations in GenTree, we incorporated complementary information from annotations of primate orthologs across 27 species, including humans (*Jovanovic et al., 2021*). This approach allowed us to assign evolutionary ages to all 337 KRAB-ZNF genes for our downstream analysis (*Figure 2—figure supplement 1*). The evolutionary age of TEs was derived from Dfam (*Storer et al., 2021*) by extracting species-specific information for each TE subfamily (*Figure 2—figure supplement 2*). Alu elements, which are primate-specific, have been extensively used in phylogenetic studies, and a subset of recently evolved Alu subfamilies is found in all Simiiformes (*Xing et al., 2007*; *Williams et al., 2010*). For downstream analyses, we classified both KRAB-ZNF genes and TEs into young and old groups based on their emergence around the divergence of Simiiformes (*Figure 2C*).

## Development of the TEKRABber software

To compare the expression of orthologous genes and TEs across species, we concatenated steps including normalization, DE, and correlation analysis into an R Bioconductor package, TEKRABber. The name was derived from the idea that TEs are bound ('grabbed') by KRAB-ZNF proteins. TEKRABber adapted the scale-based normalization method (*Zhou et al., 2019*) to normalize orthologous genes for comparison between two species. In brief, the conserved orthologous gene lengths from two species were selected from Ensembl data (*Harrison et al., 2024*) and combined with the expression data to find an optimal scaling factor that can normalize the data to achieve a minimization of deviation between empirical and nominal type I error in a hypothesis testing framework. The normalization step for TEs followed a similar concept. However, instead of using orthologous genes, we used the subset of TEs including LTR, LINE, SINE, SVAs, and DNA transposons (as defined by RepeatMasker, *Smit et al., 2013*), for which homologs can be found in other species, for scaling to normalize the expression of TEs (*Figure 1—figure supplement 1*). After normalization, differentially expressed orthologous genes and TEs were analyzed using DESeq2 v1.4.4 (*Love et al., 2014*). The one-to-one correlations between selected orthologous genes and TEs from each species were estimated. In

this analysis, we used Pearson's correlations with an adjusted p-value using the Benjamini-Hochberg correction (*Benjamini and Hochberg, 1995*) to obtain significant correlations.

## Correlation analysis

To obtain significant correlations of KRAB-ZNFs and TEs for downstream analysis, we used a workflow including the following steps. We first applied Pearson's correlations on one-to-one gene and TE, using their normalized expression levels. Then, we tested whether KRAB-ZNFs statistically had more correlations with TEs than randomly selected genes (1000 iterations, p-value<0.001). Next, we investigated the relationships between the number of correlations and correlation coefficient values. From the distribution, we decided on a threshold with an absolute value for the coefficient of larger than 0.4 and with an adjusted p-value less than 0.01. Last, we cross-validated our results by comparing the pairs found with experimentally determined TE and KRAB-ZNF relationships using ChIP-exo data from a human embryonic stem cell line (*Imbeault et al., 2017*) and testing different correlation strengths. For notation, we use TE:KRAB-ZNF to signify a correlation between a TE and a KRAB-ZNF gene, i.e., AluYc:*ZNF441* indicates a significant correlation between the expression of AluYc and *ZNF441*.

## Constructing TE:KRAB-ZNF regulatory network

Nodes and edges were selected from the TE:KRAB-ZNF results and processed into a dataframe object. The network structure was created using RCy3 v2.16 (*Gustavsen et al., 2019*) to import into Cytoscape v3.10.1 (*Shannon et al., 2003*) for visualization and editing. To visualize the regulatory networks, we applied the yFiles Organic Layout for an undirected graph by assigning nodes as objects that had repulsive or attractive forces between them (https://www.yworks.com).

## Network analysis

TE:KRAB-ZNF networks were analyzed as bipartite networks, consisting of two classes of nodes, KRAB-ZNF genes and TEs, with connections existing only between the two classes and no edges between elements from the same class. Node properties were first checked using a bipartite module in NetworkX software v2.8.4 (*Hagberg et al., 2008*). These values included bipartite degree centrality, bipartite strength centrality, and bipartite betweenness centrality for each KRAB-ZNF and TE node. The top 5% scoring genes were selected as hubs (bipartite degree and bipartite strength centrality). For cluster detection, leading eigenvector community methods (*Csárdi et al., 2023*) were used to specify unipartite community structures, and then bipartite modularity (*Barber, 2007*) was calculated with CONDOR v1.1.1 (*Platig et al., 2016*). To conduct enrichment analysis among clusters, a Fisher's exact test using 1 million simulations with a p-value<0.05 was used.

## Acknowledgements

We extend our sincere thanks to the data providers and all members of the research team involved in The Mayo RNA-seq study. We also appreciate the invaluable data resources provided (https://doi.org/10.7303/syn5550404), which have supported the progression of this research. Our sincere thanks go to Vladimir M Jovanovic, Rebecca S Saager, Jeong-Eun Költzow, Fatemeh Zebardast, Melanie Sarfert, Marula Mathew, and Vanessa H Schulmann for their contributions through critical reading and feedback.

## Additional information

### Funding

| Funder | Grant reference number | Author |
|---|---|---|
| Deutsche Forschungsgemeinschaft | NO 920/8-1 | Katja Nowick |
| Deutsche Forschungsgemeinschaft | NO 920/10-1 | Katja Nowick |

| Funder | Grant reference number | Author |
|--------|------------------------|--------|

The funders had no role in study design, data collection and interpretation, or the decision to submit the work for publication.

## Author contributions

Yao-Chung Chen, Conceptualization, Data curation, Software, Formal analysis, Validation, Visualization, Methodology, Writing – original draft, Writing – review and editing; Arnaud Maupas, Formal analysis, Visualization, Methodology, Writing – original draft, Writing – review and editing; Katja Nowick, Conceptualization, Supervision, Funding acquisition, Writing – original draft, Project administration, Writing – review and editing

## Author ORCIDs

Yao-Chung Chen https://orcid.org/0000-0002-9927-9130
Arnaud Maupas https://orcid.org/0009-0002-6471-884X
Katja Nowick https://orcid.org/0000-0003-3993-4479

## Ethics

Human subjects: This study was conducted using de-identified human sequencing data. The datasets were obtained from publicly available sources, specifically the NCBI Gene Expression Omnibus (GEO) (GSE127898) and the Synapse database (SYN5550404), for which appropriate data access approval was obtained. As the data were de-identified and no identifiable personal information was used or accessed, informed consent and ethical approval were not required for this analysis.

Reviewer #1 (Public review): https://doi.org/10.7554/eLife.103608.3.sa1
Author response https://doi.org/10.7554/eLife.103608.3.sa2

## Additional files

### Supplementary files
Supplementary file 1. Supplementary Tables.

MDAR checklist

### Data availability

The current manuscript is a computational study, so no data have been generated for this manuscript. Source code for the analysis has been uploaded to GitHub (copy archived at *Chen, 2024*).

The following previously published datasets were used:

| Author(s) | Year | Dataset title | Dataset URL | Database and Identifier |
|-----------|------|---------------|-------------|-------------------------|
| Khrameeva E, Kurochkin I, Mazin P, Khaitovich P | 2020 | Transcriptome map of the human brain at the single-cell resolution | https://www.ncbi.nlm.nih.gov/geo/query/acc.cgi?acc=GSE127898 | NCBI Gene Expression Omnibus, GSE127898 |
| SageNeuroCommunityAdmin | 2016 | Mayo RNAseq Study | https://doi.org/10.7303/syn5550404 | Synapse, 10.7303/syn5550404 |

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
