## [Editor Report · eLife Assessment]

The authors present a software (TEKRABber) to analyze how expression of transposable elements (TEs) and TE silencing factors KRAB zinc finger (KRAB-ZNF) genes are correlated in experimentally validated datasets. TEKRABber is used to reconstruct regulatory networks of KRAB-ZNFs and TEs during human brain evolution and in Alzheimer's disease. The direction of the work is **important**, with potentially significant interest from others looking for a tool for correlative gene expression analysis across individual genomes and species. However, the reviews identified biases and shortcomings in the pipeline that could lead to an unacceptable number of false positive and negative signals and thus impact the conclusions, leaving the work in its current form **incomplete**.

---

## [Referee Report · Reviewer #1 (Public review)]

The authors present their new bioinformatic tool called TEKRABber, and use it to correlate expression between KRAB ZNFs and TEs across different brain tissues, and across species. While the aims of the authors are clear and there would be significant interest from other researchers in the field for a program that can do such correlative gene expression analysis across individual genomes and species, the presented approach and work display significant shortcomings. In the current state of the analysis pipeline, the biases and shortcomings mentioned below, for which I have seen no proof of that they are accounted for by the authors, are severely impacting the presented results and conclusions. It is therefore essential that the points below are addressed, involving significant changes in the TEKRABber progamm as well as the analysis pipeline, to prevent the identification of false positive and negative signals, that would severely affect the conclusions one can raise about the analysis.

My main concerns are provided below:

One important shortcoming of the biocomputational approach is that most TEs are not actually expressed, and others (Alus) are not a proxy of the activity of the TE class at all. I will explain: While specific TE classes can act as (species-specific) promoters for genes (such as LTRs) or are expressed as TE derived transcripts (LINEs, SVAs), the majority of other older TE classes do not have such behavior and are either neutral to the genome or may have some enhancer activity as mapped in the program they refer to 'TEffectR'. A big focus is on Alus, but Alus contribute to a transcriptome in a different way too: They often become part of transcripts due to alternative splicing. As such, the presence of Alu derived transcripts is not a proxy for the expression/activity of the Alu class, but rather a result of some Alus being part of gene transcripts (see also next point). Bottom line is that the TEKRABber software/approach is heavily prone to picking up both false positives (TEs being part of transcribed loci) and false negatives (TEs not producing any transcripts at all) , which has a big implication for how reads from TEs as done in this study should be interpreted: The TE expression used to correlate the KRAB ZNF expression is simply not representing the species-specific influences of TEs where the authors are after.

With the strategy as described, a lot of TE expression is misinterpreted: TEs can be part of gene-derived transcripts due to alternative splicing (often happens for Alus) or as a result of the TE being present in an inefficiently spliced out intron (happens a lot) which leads to TE-derived reads as a result of that TE being part of that intron, rather than that TE being actively expressed. As a result, the data as analysed is not reliably indicating the expression of TEs (as the authors intend too) and should be filtered for any reads that are coming from the above scenarios: These reads have nothing to do with KRAB ZNF control, and are not representing actively expressed TEs and therefore should be removed. Given that from my lab's experience in brain (and other) tissues, the proportion of RNA sequencing reads that are actually derived from active TEs is a stark minority compared to reads derived from TEs that happen to be in any of the many transcribed loci, applying this filtering is expected to have a huge impact on the results and conclusions of this study.

Another potential problem that I don't see addressed is that due to the high level of similarity of the many hundreds of KRAB ZNF genes in primates and the reads derived from them, and the inaccurate annotations of many KZNFs in non-human genomes, the expression data derived from RNA-seq datasets cannot be simply used to plot KZNF expression values, without significant work and manual curation to safeguard proper cross species ortholog-annotation: The work of Thomas and Schneider (2011) has studied this in great detail but genome-assemblies of non-human primates tend to be highly inaccurate in appointing the right ortholog of human ZNF genes. The problem becomes even bigger when RNA-sequencing reads are analyzed: RNA-sequencing reads from a human ZNF that emerged in great apes by duplication from an older parental gene (we have a decent number of those in the human genome) may be mapped to that older parental gene in Macaque genome: So, the expression of human-specific ZNF-B, that derived from the parental ZNF-A, is likely to be compared in their DESeq to the expression of ZNF-A in Macaque RNA-seq data. In other words, without a significant amount of manual curation, the DE-seq analysis is prone to lead to false comparisons which make the stategy and KRABber software approach described highly biased and unreliable.

There is no doubt that there are differences in expression and activity of KRAB-ZNFs and TEs repspectively that may have had important evolutionary consequences. However, because all of the network analyses in this paper rely on the analyses of RNA-seq data and the processing through the TE-KRABber software with the shortcomings and potential biases that I mentioned above, I need to emphasize that the results and conclusions are likely to be significantly different if the appropriate measures are taken to get more accurate and curated TE and KRAB ZNF expression data.

Finally, there are some minor but important notes I want to share:

The association with certain variations in ZNF genes with neurological disorders such as AD, as reported in the introduction is not entirely convincing without further functional support. Such associations could be merely happen by chance, given the high number of ZNF genes in the human genome and the high chance that variations in these loci happen associate with certatin disease associated traits. So using these associations as an argument that changes in TEs and KRAB ZNF networks are important for diseases like AD should be used with much more caution.

There is a number of papers where KRAB ZNF and TE expression are analysed in parallel in human brain tissues. So the novelty of that aspect of the presented study may be limited.

Additional note after reviewing the revised version of the manuscript:

After reviewing the revised version of the manuscript, my criticism and concerns with this study are still evenly high and unchanged. To clarify, the revised version did not differ in essence from the original version; it seems that unfortunately, no efforts were taken to address the concerns raised on the original version of the manuscript, the results section as well as the discussion section are virtually unchanged.

---

## [Author Response]

The following is the authors’ response to the current reviews.

**Reviewer #1 (Public review):**
The authors present their new bioinformatic tool called TEKRABber, and use it to correlate expression between KRAB ZNFs and TEs across different brain tissues, and across species. While the aims of the authors are clear and there would be significant interest from other researchers in the field for a program that can do such correlative gene expression analysis across individual genomes and species, the presented approach and work display significant shortcomings. In the current state of the analysis pipeline, the biases and shortcomings mentioned below, for which I have seen no proof of that they are accounted for by the authors, are severely impacting the presented results and conclusions. It is therefore essential that the points below are addressed, involving significant changes in the TEKRABber progamm as well as the analysis pipeline, to prevent the identification of false positive and negative signals, that would severely affect the conclusions one can raise about the analysis.

Thank you very much for the insightful review of our manuscript. Since most of the comments on our revised version are not different from the comments on our first version, we repeated our previous answer, but wrote a new reply to the new concerns (please see the last two paragraphs).

We would also like to reiterate here that most of the critique of the reviewer concerns the performance of other tools and not TEKRABber presented in our manuscript. We consider it out of scope for this manuscript to improve other tools.

My main concerns are provided below:One important shortcoming of the biocomputational approach is that most TEs are not actually expressed, and others (Alus) are not a proxy of the activity of the TE class at all. I will explain: While specific TE classes can act as (species-specific) promoters for genes (such as LTRs) or are expressed as TE derived transcripts (LINEs, SVAs), the majority of other older TE classes do not have such behavior and are either neutral to the genome or may have some enhancer activity as mapped in the program they refer to 'TEffectR'. A big focus is on Alus, but Alus contribute to a transcriptome in a different way too: They often become part of transcripts due to alternative splicing. As such, the presence of Alu derived transcripts is not a proxy for the expression/activity of the Alu class, but rather a result of some Alus being part of gene transcripts (see also next point). Bottom line is that the TEKRABber software/approach is heavily prone to picking up both false positives (TEs being part of transcribed loci) and false negatives (TEs not producing any transcripts at all) , which has a big implication for how reads from TEs as done in this study should be interpreted: The TE expression used to correlate the KRAB ZNF expression is simply not representing the species-specific influences of TEs where the authors are after.With the strategy as described, a lot of TE expression is misinterpreted: TEs can be part of gene-derived transcripts due to alternative splicing (often happens for Alus) or as a result of the TE being present in an inefficiently spliced out intron (happens a lot) which leads to TE-derived reads as a result of that TE being part of that intron, rather than that TE being actively expressed. As a result, the data as analysed is not reliably indicating the expression of TEs (as the authors intend too) and should be filtered for any reads that are coming from the above scenarios: These reads have nothing to do with KRAB ZNF control, and are not representing actively expressed TEs and therefore should be removed. Given that from my lab's experience in brain (and other) tissues, the proportion of RNA sequencing reads that are actually derived from active TEs is a stark minority compared to reads derived from TEs that happen to be in any of the many transcribed loci, applying this filtering is expected to have a huge impact on the results and conclusions of this study.

We sincerely thank the reviewer for highlighting the potential issues of false positives and negatives in TE quantification. The reviewer provided valuable examples of how different TE classes, such as Alus, LTRs, LINEs, and SVAs, exhibit distinct behaviors in the genome. To our knowledge, specific tools like ERVmap (Tokuyama et al., 2018), which annotates ERVs, and LtrDetector (Joseph et al., 2019), which uses k-mer distributions to quantify LTRs, could indeed enhance precision by treating specific TE classes individually. We acknowledge that such approaches may yield more accurate results and appreciate the suggestion.

In our study, we used TEtranscripts (Jin et al., 2015) prior to TEKRABber. TEtranscripts applies the Expectation Maximization (EM) algorithm to assign ambiguous reads as the following steps. Uniquely mapped reads are first assigned to genes, and reads overlapping genes and TEs are assigned to TEs only if they do not uniquely match an annotated gene. The remaining ambiguous reads are distributed based on EM iterations. While this approach may not be as specialized as the latest tools for specific TE classes, it provides a general overview of TE activity. TEtranscripts outputs subfamily-level TE expression data, which we used as input for TEKRABber to perform downstream analyses such as differential expression and correlation studies.

We understand the importance of adapting tools to specific research objectives, including focusing on particular TE classes. TEKRABber is designed not to refine TE quantification at the mapping stage but to flexibly handle outputs from various TE quantification tools. It accepts raw TE counts as input in the form of dataframes, enabling diverse analytical pipelines. We would also like to clarify that, since the input data is transcriptomic, our primary focus is on expressed TEs, rather than the effects of non-expressed TEs in the genome. In the revised version of our manuscript, we emphasize this distinction in the discussion and provide examples of how TEKRABber can integrate with other tools to enhance specificity and accuracy.

Another potential problem that I don't see addressed is that due to the high level of similarity of the many hundreds of KRAB ZNF genes in primates and the reads derived from them, and the inaccurate annotations of many KZNFs in non-human genomes, the expression data derived from RNA-seq datasets cannot be simply used to plot KZNF expression values, without significant work and manual curation to safeguard proper cross species ortholog-annotation: The work of Thomas and Schneider (2011) has studied this in great detail but genome-assemblies of non-human primates tend to be highly inaccurate in appointing the right ortholog of human ZNF genes. The problem becomes even bigger when RNA-sequencing reads are analyzed: RNA-sequencing reads from a human ZNF that emerged in great apes by duplication from an older parental gene (we have a decent number of those in the human genome) may be mapped to that older parental gene in Macaque genome: So, the expression of human-specific ZNF-B, that derived from the parental ZNF-A, is likely to be compared in their DESeq to the expression of ZNF-A in Macaque RNA-seq data. In other words, without a significant amount of manual curation, the DE-seq analysis is prone to lead to false comparisons which make the stategy and KRABber software approach described highly biased and unreliable.There is no doubt that there are differences in expression and activity of KRAB-ZNFs and TEs repspectively that may have had important evolutionary consequences. However, because all of the network analyses in this paper rely on the analyses of RNA-seq data and the processing through the TE-KRABber software with the shortcomings and potential biases that I mentioned above, I need to emphasize that the results and conclusions are likely to be significantly different if the appropriate measures are taken to get more accurate and curated TE and KRAB ZNF expression data.

We thank the reviewer for raising the important issue of accurately annotating the expanded repertoire of KRAB-ZNFs in primates, particularly the challenges of cross-species orthology and potential biases in RNA-seq data analysis. Indeed, we have also addressed this challenge in some of our previous papers (Nowick et al., 2010, Nowick et al., 2011 and Jovanovic et al., 2021).

In the revised manuscript, we include more details about our two-step strategy to ensure accurate KRAB-ZNF ortholog assignments. First, we employed the Gene Order Conservation (GOC) score from Ensembl BioMart as a primary filter, selecting only one-to-one orthologs with a GOC score above 75% across primates. This threshold, recommended in Ensembl’s ortholog quality control guidelines, ensures high-confidence orthology relationships.(http://www.ensembl.org/info/genome/compara/Ortholog_qc_manual.html#goc).

Second, we incorporated data from Jovanovic et al. (2021), which independently validated KRAB-ZNF orthologs across 27 primate genomes. This additional layer of validation allowed us to refine our dataset, resulting in the identification of 337 orthologous KRAB-ZNFs for differential expression analysis (Figure S2).

We acknowledge that different annotation methods or criteria may for some genes yield variations in the identified orthologs. However, we believe that this combination provides a robust starting point for addressing the challenges raised, while we remain open to additional refinements in future analyses.

Finally, there are some minor but important notes I want to share:The association with certain variations in ZNF genes with neurological disorders such as AD, as reported in the introduction is not entirely convincing without further functional support. Such associations could be merely happen by chance, given the high number of ZNF genes in the human genome and the high chance that variations in these loci happen associate with certatin disease associated traits. So using these associations as an argument that changes in TEs and KRAB ZNF networks are important for diseases like AD should be used with much more caution.

We fully acknowledge the concern that, given the large number of KRAB-ZNFs and their inherent variability, some associations with AD or other neurological disorders could occur by chance. This highlights the importance of additional functional studies to validate the causal role of KRAB-ZNF and TE interactions in disease contexts. While previous studies have indeed analyzed KRAB-ZNF and TE expression in human brain tissues, our study seeks to expand on this foundation by incorporating interspecies comparisons across primates. This approach enabled us to identify TE:KRAB-ZNF pairs that are uniquely present in healthy human brains, which may provide insights into their potential evolutionary significance and relevance to diseases like AD.

In addition to analyzing RNA-seq data (GSE127898 and syn5550404), we have cross-validated our findings using ChIP-exo data for 159 KRAB-ZNF proteins and their TE binding regions in humans (Imbeault et al., 2017). This allowed us to identify specific binding events between KRAB-ZNF and TE pairs, providing further support for the observed associations. We agree with the reviewer that additional experimental validations, such as functional studies, are critical to further establish the role of KRAB-ZNF and TE networks in AD. We hope that future research can build upon our findings to explore these associations in greater detail.

There is a number of papers where KRAB ZNF and TE expression are analysed in parallel in human brain tissues. So the novelty of that aspect of the presented study may be limited.

We agree with the reviewer that many studies have examined the expression levels of KRAB-ZNFs and TEs in developing human brain tissues (Farmiloe et al., 2020; Turelli et al., 2020; Playfoot et al., 2021, among others). However, the novelty of our study lies in comparing KRAB-ZNF and TE expression across primate species, as well as in adult human brain tissues from both control individuals and those with Alzheimer’s disease. To our knowledge, no previous study has analyzed these data in this context. We therefore believe that our results will be of interest to evolutionary biologists and neurobiologists focusing on Alzheimer’s disease.

Additional note after reviewing the revised version of the manuscript:After reviewing the revised version of the manuscript, my criticism and concerns with this study are still evenly high and unchanged. To clarify, the revised version did not differ in essence from the original version; it seems that unfortunately, no efforts were taken to address the concerns raised on the original version of the manuscript, the results section as well as the discussion section are virtually unchanged.

We regret that this reviewer was not satisfied with our changes. In fact, many of the points raised by this reviewer are important, but concern weaknesses of other tools. In our opinion, validating other tools would be out of scope for this paper. We want to emphasize that TEKRABber is not a quantification tool for sequencing data, but a software for comparative analysis across species. We provided a detailed answer to the reviewer and readers can refer to that answer in the public review above for further information.

The following is the authors’ response to the original reviews.

**Reviewer #1 (Public review):**
The authors present their new bioinformatic tool called TEKRABber, and use it to correlate expression between KRAB ZNFs and TEs across different brain tissues, and across species. While the aims of the authors are clear and there would be significant interest from other researchers in the field for a program that can do such correlative gene expression analysis across individual genomes and species, the presented approach and work display significant shortcomings. In the current state of the analysis pipeline, the biases and shortcomings mentioned below, for which I have seen no proof that they are accounted for by the authors, are severely impacting the presented results and conclusions. It is therefore essential that the points below are addressed, involving significant changes in the TEKRABber program as well as the analysis pipeline, to prevent the identification of false positive and negative signals, that would severely affect the conclusions one can raise about the analysis.

Thank you very much for the insightful review of our manuscript.

My main concerns are provided below:(1) One important shortcoming of the biocomputational approach is that most TEs are not actually expressed, and others (Alus) are not a proxy of the activity of the TE class at all. I will explain: While specific TE classes can act as (species-specific) promoters for genes (such as LTRs) or are expressed as TE derived transcripts (LINEs, SVAs), the majority of other older TE classes do not have such behavior and are either neutral to the genome or may have some enhancer activity as mapped in the program they refer to 'TEffectR'. A big focus is on Alus, but Alus contribute to a transcriptome in a different way too: They often become part of transcripts due to alternative splicing. As such, the presence of Alu derived transcripts is not a proxy for the expression/activity of the Alu class, but rather a result of some Alus being part of gene transcripts (see also next point). The bottom line is that the TEKRABber software/approach is heavily prone to picking up both false positives (TEs being part of transcribed loci) and false negatives (TEs not producing any transcripts at all), which has a big implication for how reads from TEs as done in this study should be interpreted: The TE expression used to correlate the KRAB ZNF expression is simply not representing the species-specific influences of TEs where the authors are after.With the strategy as described, a lot of TE expression is misinterpreted: TEs can be part of gene-derived transcripts due to alternative splicing (often happens for Alus) or as a result of the TE being present in an inefficiently spliced out intron (happens a lot) which leads to TE-derived reads as a result of that TE being part of that intron, rather than that TE being actively expressed. As a result, the data as analysed is not reliably indicating the expression of TEs (as the authors intend to) and should be filtered for any reads that are coming from the above scenarios: These reads have nothing to do with KRAB ZNF control, and are not representing actively expressed TEs and therefore should be removed. Given that from my lab's experience in the brain (and other) tissues, the proportion of RNA sequencing reads that are actually derived from active TEs is a stark minority compared to reads derived from TEs that happen to be in any of the many transcribed loci, applying this filtering is expected to have a huge impact on the results and conclusions of this study.

We sincerely thank the reviewer for highlighting the potential issues of false positives and negatives in TE quantification. The reviewer provided valuable examples of how different TE classes, such as Alus, LTRs, LINEs, and SVAs, exhibit distinct behaviors in the genome. To our knowledge, specific tools like ERVmap (Tokuyama et al., 2018), which annotates ERVs, and LtrDetector (Joseph et al., 2019), which uses k-mer distributions to quantify LTRs, could indeed enhance precision by treating specific TE classes individually. We acknowledge that such approaches may yield more accurate results and appreciate the suggestion.

In our study, we used TEtranscripts (Jin et al., 2015) prior to TEKRABber. TEtranscripts applies the Expectation Maximization (EM) algorithm to assign ambiguous reads as the following steps. Uniquely mapped reads are first assigned to genes, and reads overlapping genes and TEs are assigned to TEs only if they do not uniquely match an annotated gene. The remaining ambiguous reads are distributed based on EM iterations. While this approach may not be as specialized as the latest tools for specific TE classes, it provides a general overview of TE activity. TEtranscripts outputs subfamily-level TE expression data, which we used as input for TEKRABber to perform downstream analyses such as differential expression and correlation studies.

We understand the importance of adapting tools to specific research objectives, including focusing on particular TE classes. TEKRABber is designed not to refine TE quantification at the mapping stage but to flexibly handle outputs from various TE quantification tools. It accepts raw TE counts as input in the form of dataframes, enabling diverse analytical pipelines. We would also like to clarify that, since the input data is transcriptiomic, our primary focus is on expressed TEs, rather than the effects of non-expressed TEs in the genome. In the revised version of our manuscript, we emphasize this distinction in the discussion and provide examples of how TEKRABber can integrate with other tools to enhance specificity and accuracy.

(2) Another potential problem that I don't see addressed is that due to the high level of similarity of the many hundreds of KRAB ZNF genes in primates and the reads derived from them, and the inaccurate annotations of many KZNFs in non-human genomes, the expression data derived from RNA-seq datasets cannot be simply used to plot KZNF expression values, without significant work and manual curation to safeguard proper cross species ortholog-annotation: The work of Thomas and Schneider (2011) has studied this in great detail but genome-assemblies of non-human primates tend to be highly inaccurate in appointing the right ortholog of human ZNF genes. The problem becomes even bigger when RNA-sequencing reads are analyzed: RNA-sequencing reads from a human ZNF that emerged in great apes by duplication from an older parental gene (we have a decent number of those in the human genome) may be mapped to that older parental gene in Macaque genome: So, the expression of human-specific ZNF-B, that derived from the parental ZNF-A, is likely to be compared in their DESeq to the expression of ZNF-A in Macaque RNA-seq data. In other words, without a significant amount of manual curation, the DE-seq analysis is prone to lead to false comparisons which make the strategy and KRABber software approach described highly biased and unreliable.There is no doubt that there are differences in expression and activity of KRAB-ZNFs and TEs respectively that may have had important evolutionary consequences. However, because all of the network analyses in this paper rely on the analyses of RNA-seq data and the processing through the TE-KRABber software with the shortcomings and potential biases that I mentioned above, I need to emphasize that the results and conclusions are likely to be significantly different if the appropriate measures are taken to get more accurate and curated TE and KRAB ZNF expression data.

We thank the reviewer for raising the important issue of accurately annotating the expanded repertoire of KRAB-ZNFs in primates, particularly the challenges of cross-species orthology and potential biases in RNA-seq data analysis. Indeed, we have also addressed this challenge in some of our previous papers (Nowick et al., 2010, Nowick et al., 2011 and Jovanovic et al., 2021).

In the revised manuscript, we include more details about our two-step strategy to ensure accurate KRAB-ZNF ortholog assignments. First, we employed the Gene Order Conservation (GOC) score from Ensembl BioMart as a primary filter, selecting only one-to-one orthologs with a GOC score above 75% across primates. This threshold, recommended in Ensembl’s ortholog quality control guidelines, ensures high-confidence orthology relationships. (http://www.ensembl.org/info/genome/compara/Ortholog_qc_manual.html#goc).

Second, we incorporated data from Jovanovic et al. (2021), which independently validated KRAB-ZNF orthologs across 27 primate genomes. This additional layer of validation allowed us to refine our dataset, resulting in the identification of 337 orthologous KRAB-ZNFs for differential expression analysis (Figure S2).

We acknowledge that different annotation methods or criteria may for some genes yield variations in the identified orthologs. However, we believe that this combination provides a robust starting point for addressing the challenges raised, while we remain open to additional refinements in future analyses.

(3) The association with certain variations in ZNF genes with neurological disorders such as AD, as reported in the introduction is not entirely convincing without further functional support. Such associations could merely happen by chance, given the high number of ZNF genes in the human genome and the high chance that variations in these loci happen to associate with certain disease-associated traits. So using these associations as an argument that changes in TEs and KRAB ZNF networks are important for diseases like AD should be used with much more caution.There are a number of papers where KRAB ZNF and TE expression are analysed in parallel in human brain tissues. So the novelty of that aspect of the presented study may be limited.

We fully acknowledge the concern that, given the large number of KRAB-ZNFs and their inherent variability, some associations with AD or other neurological disorders could occur by chance. This highlights the importance of additional functional studies to validate the causal role of KRAB-ZNF and TE interactions in disease contexts. While previous studies have indeed analyzed KRAB-ZNF and TE expression in human brain tissues, our study seeks to expand on this foundation by incorporating interspecies comparisons across primates. This approach enabled us to identify TE:KRAB-ZNF pairs that are uniquely present in healthy human brains, which may provide insights into their potential evolutionary significance and relevance to diseases like AD.

In addition to analyzing RNA-seq data (GSE127898 and syn5550404), we have cross-validated our findings using ChIP-exo data for 159 KRAB-ZNF proteins and their TE binding regions in humans (Imbeault et al., 2017). This allowed us to identify specific binding events between KRAB-ZNF and TE pairs, providing further support for the observed associations. We agree with the reviewer that additional experimental validations, such as functional studies, are critical to further establish the role of KRAB-ZNF and TE networks in AD. We hope that future research can build upon our findings to explore these associations in greater detail.

**Reviewer #1 (Recommendations for the authors):**
It is essential before this work can be considered for publication, that the points above are addressed, involving significant changes in the TEKRABber program as well as the analysis pipeline, to prevent the identification of false positive and negative signals, that would severely affect the conclusions one can raise about the analysis.

We sincerely appreciate the reviewer’s insightful recommendations and constructive feedback. Each specific point has been carefully addressed in detail in the public reviews section above.

**Reviewer #2 (Public review)**
Summary:The aim was to decipher the regulatory networks of KRAB-ZNFs and TEs that have changed during human brain evolution and in Alzheimer's disease.Strengths:This solid study presents a valuable analysis and successfully confirms previous assumptions, but also goes beyond the current state of the art.Weaknesses:The design of the analysis needs to be slightly modified and a more in-depth analysis of the positive correlation cases would be beneficial. Some of the conclusions need to be reinterpreted.

We sincerely thank the reviewer for the thoughtful summary, positive evaluation of our study, and constructive feedback. We appreciate the recognition of the strengths in our analysis and the valuable suggestions for improving its design and interpretation.

We would like to briefly comment on the suggested modifications to the design here and will provide a detailed point-by-point review later with our revised manuscript.

The reviewer recommended considering a more recent timepoint, such as less than 25 million years ago (mya), to define the "evolutionary young group" of KRAB-ZNF genes and TEs when discussing the arms-race theory. This is indeed a valuable perspective, as the TE repressing functions by KRAB-ZNF proteins may have evolved more recently than the split between Old World Monkeys (OWM) and New World Monkeys (NWM) at 44.2 mya we used.

Our rationale for selecting 44.2 mya is based on certain primate-specific TEs such as the Alu subfamilies, which emerged after the rise of Simiiformes and have been used in phylogenetic studies (Xing et al., 2007 and Williams et al., 2010). This timeframe allowed us to investigate the potential co-evolution of KRAB-ZNFs and TEs in species that emerged after the OWM-NWM split (e.g., humans, chimpanzees, bonobos, and macaques used for this study). However, focusing only on KRAB-ZNFs and TEs younger than 25 million years would limit the analysis to just 9 KRAB-ZNFs and 92 TEs expressed in our datasets. While we will not conduct a reanalysis using this more recent timepoint, we will integrate the recommendation into the discussion section of the revised manuscript.

Furthermore, we greatly appreciate the reviewer's detailed insights and suggestions for refining specific descriptions and interpretations in our manuscript. We will address these points in the revised version to ensure the content is presented with greater precision and clarity.

Once again, we thank both reviewers for their valuable feedback, which provides significant input for strengthening our study.

**Reviewer #2 (Recommendations for the authors):**
We thank the reviewer for the very insightful comments, which helped a lot in our interpretation and discussion of our results and in improving some of our statements.The present study seeks to uncover how the repression of transposable elements (TEs) by rapidly evolving KRAB-ZNF genes, which are known for their role in TE suppression, may influence human brain evolution and contribute to Alzheimer's disease (AD). Utilizing their previously developed tool, TEKRABber, the researchers analyze transcriptome datasets from the brains of four species of Old World Monkeys (OWM) alongside samples from healthy human individuals and AD patients.Through bipartite network analysis, they identify KRAB-ZNF/Alu-TE interactions as the most negatively correlated in the network, highlighting the repression of Alu elements by KRAB-ZNF proteins. In AD patient samples, they observe a reduction in a subnetwork comprising 21 interactions within an Alu TE module. These findings are consistent with earlier evidence that: (1) KRAB-ZNFs are involved in suppressing evolutionarily young Alu TEs; and (2) specific Alu elements have been reported to be deregulated in AD. The study also validates previous experimental ChIP-exo data on KRAB-ZNF proteins obtained in a different cell type (Imbeault et al., 2017).As a novely, the study identifies a human-specific amino acid variation in ZNF528, which directly contacts DNA nucleotides, showing signs of positive selection in humans and several human-specific TE interactions.Interestingly, in addition to the negative links, the researchers observed predominantly positive connections with other TEs, suggesting that while their approach is consistent with some previous observations, the authors conclude that it provides limited support for the 'genetic arms race' hypothesis.The reviewer is a specialist in TE and evolutionary research.Major issues:The study demonstrates the usefulness of the TEKRABber tool, which can support and successfully validate previous observations. However, there are several misconceptions and problems with the interpretation of the results.KRAB-ZNF proteins in repressing TEs in vertebrates In the Abstract: "In vertebrates, some KRAB-ZNF proteins repress TEs, offering genomic protection."Although some KRAB-ZNF proteins exist in vertebrates, their TE-suppression role is not as prominent or specialized as it is in mammals, where it serves as a key defense mechanism against the mobilization of TEs.

We appreciate the reviewer’s clarification regarding the role of KRAB-ZNF proteins in vertebrates. To improve accuracy and precision, we have revised the wording to specify that this mechanism is primarily observed in mammals rather than vertebrates.

The definition of young and oldThe study considers the evolutionary age of young ({less than or equal to} 44.2 mya) and old(> 44.2 mya). This is the time of the Old World Monkey (OWM) and New World Monkey (NWM) split. Importantly, however, the KRAB-ZNF / KAP1 suppression system primarily suppresses evolutionarily younger TEs (< 25 MY old). These TEs are relatively new additions to the genome, i.e. they are specific to certain lineages (such as primates or hominins) and are more likely to be actively transcribed (and recognized as foreign by innate immunity) or have residual activity upon transposition. Examples include certain subfamilies of LINE-1, Alu (Y, S, less effective for J), SVA and younger human endogenous retroviruses (HERVs) such as HERV-K. The KRAB-ZNF / KAP1 system therefore focuses primarily on TEs that have evolved more recently in primates, in the last few million years (within the last 25 million years). Older TEs are controlled by broader epigenetic mechanisms such as DNA methylation, histone modifications, etc. Therefore, the age ({less than or equal to} 44.2 mya) is not suitable to define it as young.In this context, the specific TEs of the Simiiformes cannot be considered as 'recently evolved' (in the Abstract). The Simiiformes contain both OWM and NWM. Notably, the study includes four species, all of which belong to the OWMs.The 'genetic arms race' theoryUnfortunately, the problematic definition of young and old could also explain why the authors conclude that their data only weakly support the 'genetic arms race' hypothesis.The KRAB-ZNF proteins evolve rapidly, similar to TEs, which raises the 'genetic arms race' hypothesis. This hypothesis refers to the constant evolutionary struggle between organisms and TEs. TEs constantly evolve to overcome host defences, while host genomes develop mechanisms to suppress these potentially harmful elements. Indeed, in mammals, an important example is the KRAB-ZNF/TE interaction. The KRAB-ZNF proteins rapidly evolve to target specific TEs, creating a 'genetic arms race' in which each side - TEs and the KRAB-ZNF/KAP1 (alias TRIM28) repressor complex - drives the evolution of the other in response to adaptive pressure. Importantly, the 'genetic arms race' hypothesis describes the evolutionary process that occurs between TE and host when the TE is deleterious. Again, this includes the young TEs (< 25 MY old) with residual transposition activity or those that actively transcribed and exacerbate cellular stress and inflammatory responses. Approximately 25 million years ago, the superfamilies Hominoidea (apes) and Cercopithecoidea (Old World monkeys, I.e. macaque) split.

Just to clarify, our initial study aim was to examine whether TEs exhibit any evolutionary relationships with KRAB-ZNFs across the four studied species (human, chimpanzee, bonobo, and macaque). For investigating the arms-race hypothesis, we really appreciate the reviewer suggesting a more recent time point, such as less than 25 million years ago (mya), to define the "evolutionary young group" of TEs and KRAB-ZNF genes. This is indeed a valuable recommendation, as 25 mya marks the emergence of Hominoidea (Figure 2C in the manuscript), making it a meaningful reference point for studying recently evolved KRAB-ZNFs and TEs. However, restricting the analysis to elements younger than 25 mya would reduce the dataset to only 9 KRAB-ZNFs and 92 TEs. Nevertheless, we provide here our results for those elements in Table S7:

We observed that among the correlations in the < 25 mya subset, negative correlations (7) outnumbered positive ones (2). However, these correlations were derived from only 3 out of 9 KRAB-ZNFs and 9 out of 92 TE subfamilies. Therefore, based on our data, while the < 25 mya group shows a higher proportion of negative correlations, the sample size is too limited to derive networks or draw robust conclusions in our analysis, especially when compared to our original evolutionary age threshold of 44.2 mya. For this reason, we chose not to reanalyze the data but rather to acknowledge that our current definition of “young” may not be optimal for testing the arms-race model in humans. While previous studies (Jacobs et al., 2014; Bruno et al., 2019; Zuo et al., 2023) have explored relevant KRAB-ZNF and TE interactions, our review of the KRAB-ZNFs and TEs highlighted in those works suggests that a specific focus on elements <25 mya has not been a primary emphasis.

"our findings only weakly support the arms-race hypothesis. Firstly, we noted that young TEs exhibit lower expression levels than old TEs (Figure 2D and 5B), which might not be expected if they had recently escaped repression". - This is a misinterpretation. These old TEs are no longer harmful. This is not the case of the 'genetic arms race'.

We sincerely appreciate the reviewer’s comments, which have helped us refine our interpretation to prevent potential misunderstandings. Our initial expectation, based on the arms-race hypothesis, was that young TEs would exhibit higher expression levels due to a recent escape from repression, while young KRAB-ZNFs would show increased expression as a counter-adaptive response. However, our findings indicate that both young TEs and young KRAB-ZNFs exhibit lower expression levels. This observation does not align with the classical arms-race model, which typically predicts an ongoing cycle of adaptive upregulation. We rephrase the sentences in our discussion to hopefully make our idea more clear. In addition, we added the notion that older TEs might not be harmful anymore, which we agree with.

"Additionally, some young TEs were also negatively correlated with old KRAB-ZNF genes, leading to weak assortativity regarding age inference, which would also not be in line with the arms-race idea."This is not a contradiction, as an old KRAB-ZNF gene could be 'reactivated' to protect against young TEs. It might be cheaper for the host than developing a brand new KRAB-ZNF gene.

We agree with the reviewer's point that older KRAB-ZNFs may be reactivated to suppress young TEs, potentially as a more cost-effective evolutionary strategy than the emergence of entirely new KRAB-ZNFs. We have incorporated this perspective into the revised manuscript to provide a more detailed discussion of our findings.

TEs remain activeIn the abstract: "Notably, KRAB-ZNF genes evolve rapidly and exhibit diverse expression patterns in primate brains, where TEs remain active."This is not precise. TEs are not generally remain active in the brain. It is only the autonomous LINE-1 (young) and non-autonomous Alu (young) and SVA (young) elements that can be mobilized by LINE-1. In addition, the evolutionary young HERV-K is recognized as foreign and alerts the innate immune system (DOI: 10.1172/jci.insight.131093) and is a target of the KRAB-ZNF/KAP1 suppression system.In the abstract: "Evidence indicates that transposable elements (TEs) can contribute to the evolution of new traits, despite often being considered deleterious."Oversimplification: The harmful and repurposed TEs are washed together.

We appreciate the reviewer’s detailed suggestions for improving the precision of our abstract. While we previously mentioned LINE-1 and Alu elements in the introduction, we now explicitly specify in the abstract that only certain TE subfamilies, such as autonomous LINE-1 and non-autonomous Alu and SVA elements, remain active in the primate brain. Additionally, we have refined the phrasing regarding the role of TEs in evolution to clearly distinguish between their deleterious effects and their potential for functional repurposing. These clarifications have been incorporated into the revised abstract to ensure greater accuracy and nuance.

Positive links"The high number of positive correlations might be surprising, given that KRAB-ZNFs are considered to repress TEs."Based on the above, it is not surprising that negative associations are only found with young (< 25 my) TEs. In fact, the relationship between old KRAB-ZNF proteins and old (non-damaging) TEs could be neutral/positive. The case of ZNF528 could be a valuable example of this.

We thank the reviewer for providing this plausible interpretation and added it to the manuscript.

"276 TE:KRAB-ZNF with positive correlations in humans were negatively correlated in bonobos" It would be important to characterise the positive correlations in more detail. Could it be that the old KRAB-ZNF proteins lost their ability to recruit KAP1/TRIM28? Demonstrate it.The strategy of developing sequence-specific DNA recognition domains that can specifically recognise TEs is expensive for the host. Recent studies suggest that when the TE is no longer harmful, these proteins/connections can be occasionally repurposed. The repurposed function would probably differ from the original suppressive function.In my opinion, the TEKRABber tool could be useful in identifying co-option events:

We appreciate the reviewer’s suggestion regarding the characterization of positive correlations. While it is possible that some old KRAB-ZNF proteins have lost their ability to recruit KAP1/TRIM28, we cannot conclude this definitively for all cases. To address this, we examined ChIP-exo data from Imbeault et al. (2017) (Accession: GSE78099) and analyzed the overlap of binding sites between KRAB-ZNFs, KAP1/TRIM28, and RepeatMasker-annotated TEs. Our results indicate that some old KRAB-ZNFs still exhibit binding overlap with KAP1 at TE regions, suggesting that their repressive function may be at least partially retained (Author response image 1).

**Author response image 1. sa2fig1:** Overlap of KAP1, Zinc finger proteins, and RepeatMasker annotation. Here we detect the overlap of ChIP-exo binding events using KAP1/TRIM28, with KRAB-ZNF genes (one at a time) and RepeatMasker annotation. (115 old and 58 young KRAB-ZNFs, Mann-Whitney, p<0.01).

Minor"Lead poisoning causes lead ions to compete with zinc ions in zinc finger proteins, affecting proteins such as DNMT1, which are related to the progression of AD (Ordemann and Austin 2016)."Not precise: While DNMT1 does contain zinc-binding domains, it is not categorized as a zinc finger protein.

We appreciate the reviewer’s insight regarding the classification of DNMT1. After careful consideration, we have removed this sentence from the introduction to maintain focus on KRAB zinc finger proteins.

Definition of TEs"There were 324 KRAB-ZNFs and 895 TEs expressed in Primate Brain Data." Define it more precisely. It is not clear, what the authors mean by TEs: Are these TE families, subfamilies? Provide information on copy numbers of each in the analysed four species.

We appreciate the reviewer’s suggestion to clarify our definition of TEs. To improve precision, we have specified that the analysis was conducted at the subfamily level. Additionally, we have provided the copy numbers of TEs for the four analyzed species in Table S4.

Occupancy of TEs in the genome"TEs comprise (i) one third to one half of the mammalian genome and are (ii) not randomly distributed..."(i) The most accepted number is 45%. However, some more recent reports estimate over 50%, thus the one third is an underestimation.(ii) Not randomly distributed among the mammalian species?

(i) We thank the reviewer for pointing out that our statement about the abundance of TEs was outdated. We have updated the estimate to reflect that TEs can occupy more than half of the genome, based on recent publications.

(ii) We acknowledge the reviewer’s concern regarding the distribution of TEs. Although TEs are interspersed throughout the genome, their insertion sites are not entirely random, as they tend to exhibit preferences for certain genomic regions. To clarify this, we have revised the wording in the paragraph accordingly.

We would like to express our sincere gratitude to both reviewers for their insightful feedback, which has been instrumental in enhancing the quality of our study.